# ReForm: Reflective Autoformalization with Prospective Bounded Sequence Optimization

**Guoxin Chen**[1,4,*], **Jing Wu**[2,*,†], **Xinjie Chen**[2,3], **Wayne Xin Zhao**[1,4,†], **Ruihua Song**[1,†]
**Chengxi Li**[2], **Kai Fan**[2], **Dayiheng Liu**[2], **Minpeng Liao**[2,†]
[1]Gaoling School of Artificial Intelligence, Renmin University of China
[2]Tongyi Lab, Alibaba Group, [3]Zhejiang University
[4]Beijing Key Laboratory of Research on Large Models and Intelligent Governance
{gx.chen.chn, batmanfly}@gmail.com, songruihua_bloon@outlook.com
{lz.wujing, chenxinjie.cxj, minpeng.lmp}@alibaba-inc.com
🤗 Model 🗄 Benchmark ⓞ Code

## Abstract

Autoformalization, which translates natural language mathematics into machine-verifiable formal statements, is critical for using formal mathematical reasoning to solve math problems stated in natural language. While Large Language Models can generate syntactically correct formal statements, they often fail to preserve the original problem's semantic intent. This limitation arises from the LLM approaches' treating autoformalization as a simplistic translation task which lacks mechanisms for self-reflection and iterative refinement that human experts naturally employ. To address these issues, we propose ReForm, a Reflective Autoformalization method that tightly integrates semantic consistency evaluation into the autoformalization process. This enables the model to iteratively generate formal statements, assess its semantic fidelity, and self-correct identified errors through progressive refinement. To effectively train this reflective model, we introduce Prospective Bounded Sequence Optimization (PBSO), which employs different rewards at different sequence positions to ensure that the model develops both accurate autoformalization and correct semantic validations, preventing superficial critiques that would undermine the purpose of reflection. Extensive experiments across four autoformalization benchmarks demonstrate that ReForm achieves an average improvement of 22.6 percentage points over the strongest baselines. To further ensure evaluation reliability, we introduce ConsistencyCheck, a benchmark of 859 expert-annotated items that not only validates LLMs as judges but also reveals that autoformalization is inherently difficult: even human experts produce semantic errors in up to 38.5% of cases.

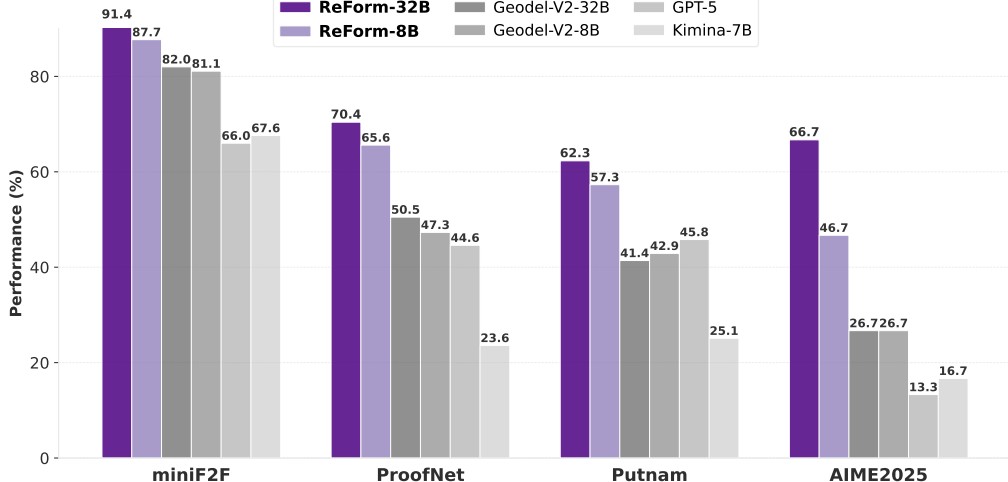

Figure 1: Autoformalization performance of ReForm against state-of-the-art models.

---

* Equal Contribution. † Corresponding Authors.

# 1 INTRODUCTION

Recent advances in Formal Mathematical Reasoning have demonstrated remarkable capabilities across a variety of challenging scenarios (Polu & Sutskever, 2020; Yang et al., 2023; Xin et al., 2024a;b; InternLM Team, 2023; Wu et al., 2024; Li et al., 2024; Wu et al., 2025; Ren et al., 2025; Lin et al., 2025a; Weng et al., 2025). However, these advances are unevenly distributed across two symbiotic tasks: Automated Theorem Proving (ATP), the process of finding a proof for a given formal statement, and Autoformalization, the translation of natural language mathematical problems into formal, machine-verifiable statements such as those in Lean (De Moura et al., 2015). This disparity establishes autoformalization as a critical bottleneck: it remains a labor-intensive endeavor that poses a great challenge even for human experts (Chen et al., 2025b).

While Large Language Models (LLMs) (Hurst et al., 2024; Anthropic, 2025; Yang et al., 2025; Google, 2025) have shown proficiency in generating syntactically well-formed statements that pass Lean compiler verification (**syntactic correctness**), they often struggle to faithfully capture the semantic intent of the original problem (**semantic consistency**). Recent studies (Peng et al., 2025) have highlighted the pervasive issue of poor semantic fidelity in the current autoformalization systems. Several concurrent works (Wang et al., 2025a; Lin et al., 2025b) have attempted to address this challenge by curating high-quality datasets specifically designed to improve semantic consistency while still treating autoformalization as a direct translation task in which models generate formal statements in a single forward pass, an approach we term the *one-pass generation paradigm*. While these data-centric efforts yield notable improvements, we find that models trained under this paradigm still frequently fail on subtle semantic details, such as misinterpreting quantifier scopes, overlooking implicit constraints, incorrectly formalizing edge cases, etc., that fundamentally compromise the original problem's intended meaning. In this work, we argue that the root of this persistent limitation lies not only in the data quality, but more fundamentally in the one-pass generation paradigm itself: without any mechanism for self-reflection and correction, models cannot progressively identify and resolve their own semantic errors during generation. This stands in stark contrast to how human experts tackle autoformalization. They employ an iterative process of review and refinement, continuously validating and adjusting their formal statements to ensure semantic fidelity.

Inspired by this, we propose REFORM, a novel *Reflective Autoformalization paradigm* that emulates the human process of iterative review and refinement to enhance semantic consistency. Instead of treating autoformalization as a single-pass translation task, REFORM reconceptualizes it as a reflective, iterative process that interweaves autoformalization with semantic self-validation. Specifically, REFORM operates through a self-correction loop: (1) it first generates a candidate formal statement, (2) then critically evaluates whether this formalization faithfully captures the original problem's semantics, and (3) iteratively refines the statement based on the identified semantic discrepancies. Unlike traditional one-pass approaches that commit to a single translation, this reflective paradigm enables the model to detect and correct its own semantic errors during the generation process, significantly reducing the risk of meaning distortions.

To prevent superficial or hallucinated critiques in the semantic self-validation, we design a *heterogeneous reward mechanism* that targets two synergistic objectives: achieving correct final formal statements as the primary task ($r_{\text{task}}$ rewarded at the sequence end) and producing accurate semantic validation critiques as the auxiliary task ($r_{\text{aux}}$ rewarded at intermediate steps). However, optimizing such heterogeneous rewards poses a significant challenge for existing reinforcement learning (RL) methods, which typically handle only a single terminal reward. We therefore introduce *Prospective Bounded Sequence Optimization (PBSO)*, a novel RL algorithm that enables optimizing multiple reward signals at different sequence positions. The key innovation of PBSO lies in its *prospective bounded return*, which smoothly integrates these heterogeneous signals by computing a discounted sum of future rewards for each step, while crucially bounding these returns within the reward function's range to prevent unbounded accumulation and ensure training stability. This enables effective credit assignment across steps with different reward objectives. That is the validation steps learn to produce accurate critiques that facilitate later corrections, while generation steps benefit from the improved validation signals. By optimizing these complementary objectives within each sequence, the model both develops stronger self-validation capabilities and achieves better autoformalization performance, with each capability reinforcing the other throughout the training process.

Extensive experiments across four challenging autoformalization benchmarks validate the effectiveness of our REFORM. We achieve an average improvement of 22.6 percentage points over the

strongest baselines. In the choice of evaluation metrics, while recent works predominantly rely on LLMs as judges for semantic consistency evaluation in autoformalization (Wang et al., 2025a; Lin et al., 2025b), the reliability of these LLM-based judges is not sufficiently studied. To rigorously investigate the reliability of frontier LLMs as evaluation metrics, we construct **ConsistencyCheck**, a benchmark of 859 expert-annotated items to test how accurately a model determines whether a given formal statement correctly captures the problem's intent. Our analysis on ConsistencyCheck reveals three insights: (1) **Human Expert Fallibility**: 16.4% of miniF2F and 38.5% of ProofNet's human-written formal statements contain semantic errors, demonstrating that autoformalization challenges even human experts. (2) **Evaluation Reliability Despite Imperfection**: Frontier LLMs make correct determination 85.8% of the time, indicating sufficient reliability as an evaluation metric. Crucially, REFORM's substantial improvements far exceed the potential evaluation noise, confirming the robustness of our findings. (3) **Classification-Generation Gap**: This binary classification task is conceptually simpler than autoformalization task. However, its maximum performance is 85.8%, which helps explain why generating semantically faithful formalizations remains stubbornly difficult. These results confirm the effectiveness of our reflective method in producing more reliable and semantically faithful autoformalization.

In summary, our contributions are as follows:

- We propose REFORM, a reflective autoformalization paradigm that reconceptualizes autoformalization from one-pass translation to an iterative process interweaving generation with semantic self-validation, enabling progressive error identification and correction.
- We introduce Prospective Bounded Sequence Optimization (PBSO) to handle different reward signals at different sequence positions through prospective bounded returns that improve both autoformalization and semantic validation.
- We construct ConsistencyCheck benchmark to rigorously evaluate the reliability of LLM-based metrics and the quantify the challenges autoformalization poses.
- We demonstrate REFORM's effectiveness across four challenging benchmarks. The model achieves an average improvement of 22.6% while maintaining computational efficiency.

## 2 RELATED WORKS

**Autoformalization.** Autoformalization—the translation of natural language problems into machine-verifiable formal languages—plays a pivotal role in formal mathematical reasoning (Wu et al., 2023; Jiang et al., 2023; Ying et al., 2024; Xie et al., 2025; He et al., 2025; Zhang et al., 2025; Jiayi et al., 2025; Yu et al., 2025b). While early approaches achieved syntactic correctness through increasing training data (Han et al., 2024; Xin et al., 2024b), recent studies (Peng et al., 2025) reveal that these models suffer from pervasive semantic infidelity. Recent concurrent works (Wang et al., 2025a; Lin et al., 2025b) focus on semantically-enhanced datasets to address this issue, yet they remain constrained by the *one-pass generation paradigm* without correction mechanisms. In contrast, our REFORM interweaves autoformalization with semantic self-validation, allowing the model to progressively identify and correct its own semantic errors during generation.

**Reinforcement Learning for LLM Reasoning.** Reinforcement Learning (RL) has emerged as a powerful paradigm for enhancing LLM reasoning capabilities (Schulman et al., 2017; Shao et al., 2024; Chen et al., 2024a;b; Wang et al., 2025b; Yu et al., 2025a; Qiao et al., 2025; Chen et al., 2025a). However, existing methods predominantly rely on terminal-only rewards. While effective for single-objective tasks, this reward paradigm fails to monitor intermediate validation steps in multi-objective tasks, leading to superficial or hallucinated critiques that undermine the self-correction process. In contrast, our PBSO introduces a *prospective bounded return* to integrate heterogeneous rewards across sequence positions. This enables position-specific optimization for both reflective autoformalization and general sequential decision-making tasks with multi-objective requirements.

## 3 METHODOLOGY

As illustrated in Figure 2, we present REFORM, our reflective autoformalization framework that departs from the prevailing one-pass generation paradigm by introducing an iterative self-correction process for enhanced semantic consistency. In this section, we describe the Reflective Autoformalization Paradigm (§3.1) followed by the Prospective Bounded Sequence Optimization (§3.2).

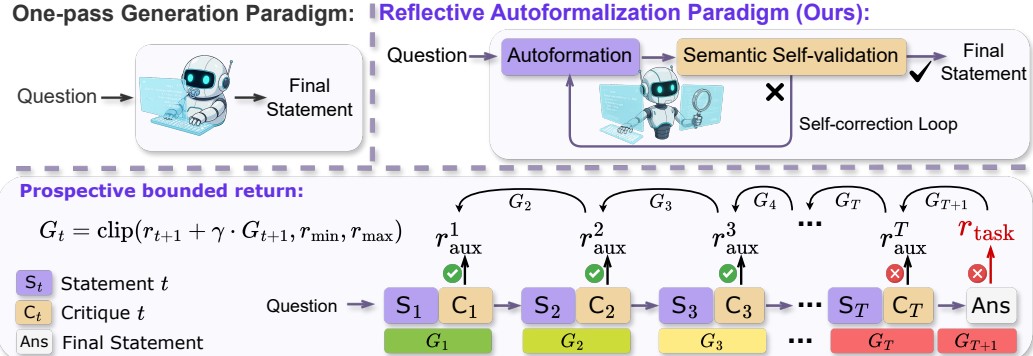

Figure 2: Overview of REFORM. **(Top)** Unlike traditional one-pass generation, our REFORM reconceptualizes it as a iterative process that interweaves autoformalization with semantic self-validation. **(Bottom)** We assign heterogeneous rewards across iterations: auxiliary rewards $r_{\text{aux}}^t$ for critique quality and task reward $r_{\text{task}}$ for final correctness. Prospective bounded returns $G_t$ computed through clipped backward accumulation enable fine-grained credit assignment for each iteration, preventing the degeneration of self-validation while improving autoformalization performance.

## 3.1 REFLECTIVE AUTOFORMALIZATION PARADIGM

The core innovation of REFORM lies in reconceptualizing autoformalization as an iterative refinement process that interweaves formal statement generation with semantic self-validation. Unlike traditional one-pass approaches, our reflective paradigm establishes a self-correction loop where the model progressively refines its output based on its own semantic critiques.

Given a natural language mathematical question $Q$, REFORM operates through a sequence of refinement iterations. At iteration $t$, the model maintains the complete history of previous attempts: $\mathcal{H}_t = \{(S_1, C_1), \ldots, (S_{t-1}, C_{t-1})\}$, where $S_j$ and $C_j$ denote the $j$-th formal statement and its corresponding semantic critique. This history enables the model to learn from its previous attempts, avoiding repeated errors and progressively converging toward a semantically faithful formalization. Each iteration comprises two interconnected stages:

1. **Autoformalization:** The model $\pi$ generates a new formal statement $S_t$ conditioned on the question $Q$ and history $\mathcal{H}_t$. For the initial iteration ($t = 1$), the model performs standard autoformalization: $S_1 = \pi(Q)$. For subsequent iterations ($t > 1$), the generation leverages insights from previous critiques to address identified semantic issues: $S_t = \pi(Q, \mathcal{H}_t)$.

2. **Semantic Self-Validation:** Given the newly generated statement $S_t$, the model produces a critique $C_t = \pi(Q, \mathcal{H}_t, S_t)$ that assesses the semantic consistency between $S_t$ and $Q$. The critique $C_t$ provides detailed diagnostic feedback, identifying specific semantic discrepancies, enabling targeted improvements in subsequent iterations.

This "Autoformalization ↔ Self-validation" loop continues until the critique indicates satisfactory semantic fidelity after which the model produces the final statement.

**Unified Generation.** While conceptually iterative, our paradigm is implemented as a single continuous autoregressive generation. Both autoformalization and self-validation naturally unfold within the same generation sequence: the model generates $S_t$, which becomes part of the context for generating $C_t$, which in turn informs $S_{t+1}$. This design ensures that the entire reflective process occurs within a single forward pass, maintaining computational efficiency comparable to the one-pass methods while achieving superior semantic consistency. The model learns to autonomously identify and correct semantic errors during generation, effectively internalizing what would traditionally require multiple model calls and human oversight.

**Mutual Reinforcement with Dual Capabilities.** The interweaving of autoformalization and self-validation creates a virtuous cycle: the autoformalization capability progressively develops semantic awareness by learning from self-validation, while the self-validation capability becomes increasingly adept at identifying subtle errors by observing refinement patterns. This mutual reinforcement stands

in contrast to traditional one-pass approaches and is instrumental in enabling REFORM to achieve formal statements that are both syntactically valid and semantically faithful.

## 3.2 PROSPECTIVE BOUNDED SEQUENCE OPTIMIZATION

While the reflective autoformalization paradigm establishes the structural framework for iterative refinement, training models to effectively execute this process presents a unique challenge: *how to jointly optimize for both correct final autoformalizations and accurate intermediate self-validations within a single sequence?*

The core difficulty lies in the heterogeneous nature of these objectives. High-quality self-validation critiques are essential for guiding refinement, yet they serve as diagnostic tools rather than direct solutions. A model might generate insightful critiques identifying all semantic issues but fail to translate these insights into correct refinements, or conversely, produce correct formalizations despite superficial self-validations. This creates an important credit assignment problem: optimizing solely for final task success provides no explicit signal for the critique quality, potentially causing the self-validation mechanism to degenerate into trivial or hallucinated assessments.

### 3.2.1 HETEROGENEOUS REWARD MECHANISM

To address this challenge, we introduce a heterogeneous reward structure that supervises both the primary autoformalization task and the auxiliary self-validation task:

**Task Reward for Autoformalization.** We assign a positive reward to the final formal statement (termed '**Ans**' below) only when it achieves both syntactic and semantic correctness:

$$r_{\text{task}}(Q, \text{Ans}) = \begin{cases} 1 & \text{if } \texttt{PassesLean}(\text{Ans}) \wedge \texttt{IsConsistent}(Q, \text{Ans}) \\ 0 & \text{otherwise} \end{cases} \quad (1)$$

where `PassesLean` verifies syntactic validity through the Lean compiler, and `IsConsistent` assesses semantic consistency between "Ans" and the original question $Q$ using an LLM-based judge[1]. This reward drives the primary learning objective, encouraging the model to produce correct final formalizations.

**Auxiliary Rewards for Self-Validation Quality.** To prevent degeneration of self-validation, we introduce auxiliary rewards that directly supervise each critique $C_t$:

$$r_{\text{aux}}^t(Q, S_t, C_t) = \begin{cases} 1 & \text{if } \texttt{IsFaithfulCritique}(Q, S_t, C_t) \\ 0 & \text{otherwise} \end{cases} \quad (2)$$

where `IsFaithfulCritique` evaluates whether $C_t$ accurately diagnoses the semantic relationship between the current statement $S_t$ and question $Q$, penalizing false positives, false negatives, and premature termination (incorrectly claiming semantic fidelity when discrepancies remain).

Together, these heterogeneous rewards provide complementary supervision signals. The task reward ensures correct final outputs while auxiliary rewards maintain the integrity of the self-validation mechanism, jointly enabling effective reflective autoformalization.

### 3.2.2 PROSPECTIVE BOUNDED RETURN

Existing RL methods typically assign rewards only at sequence termination, optimizing solely for task success. In our reflective paradigm, this method would provide little to no supervision for the quality of intermediate self-validations. Without explicit rewards for self-validation, the model is at risk of learning to generate superficial or hallucinated critiques that appear to justify refinements but provide no genuine diagnostic value, thereby undermining the entire reflective mechanism.

To address this problem, we introduce a prospective bounded return that maximizes expected cumulative reward while ensuring quality at each step. Our approach integrates both task and auxiliary rewards distributed across the trajectory, where each position's return $G_t$ captures the cumulative value of the remaining sequence from that point forward. This prospective view enables the model

---

[1] `IsConsistent` and `IsFaithfulCritique` are evaluated by `CriticLean-14B` (Peng et al., 2025) and `Qwen3-235B-A22B` (Yang et al., 2025), respectively. We provide detailed reliability evaluations in § 4.5.

to learn how current decisions contribute to eventual task success. For a trajectory with $T$ iterations producing rewards $[r_{\text{aux}}^1, \ldots, r_{\text{aux}}^T, r_{\text{task}}]$, we compute returns for each step through backward accumulation with bounded discounting:

$$G_t = \text{clip}\left(r_t + \gamma \cdot G_{t+1}, r_{\min}, r_{\max}\right) \tag{3}$$

where $\gamma \in (0, 1]$ is the discount factor, $G_{T+1} = 0$, and the clipping operation bounds returns within the reward function's range $[r_{\min}, r_{\max}]$ to prevent gradient instability from unbounded accumulation. Each $G_t$ serves as the composite reward signal for the entire $t$-th iteration—encompassing both the statement generation $S_t$ and its critique $C_t$—capturing how this complete reflective step contributes to the trajectory's overall success.

### 3.2.3 SEQUENCE OPTIMIZATION WITH POSITION-SPECIFIC ADVANTAGES

Building on the prospective bounded returns, we now present our complete Prospective Bounded Sequence Optimization (PBSO) algorithm. Unlike existing RL methods that compute advantages using only terminal task rewards without supervising intermediate steps, PBSO leverages the full sequence of heterogeneous returns to compute position-specific advantages. This enables fine-grained credit assignment where each iteration receives distinct supervision based on its actual contribution to the trajectory's success.

For each question $Q$, we sample $N$ complete trajectories, where trajectory $j$ undergoes $T_j$ iterations. The bounded return computation (Eq. 3) yields a sequence of returns $\{G_1^j, G_2^j, \ldots, G_{T_j+1}^j\}$ capturing the prospective value at each iteration. To enable policy optimization, we transform these returns into advantages through joint normalization across all sampled trajectories:

$$\hat{A}_t^j = \frac{G_t^j - \text{mean}(\mathcal{G})}{\text{std}(\mathcal{G})}, \quad \text{where} \quad \mathcal{G} = \bigcup_{j=1}^{N} \{G_t^j : t = 1, \ldots, T_j + 1\} \tag{4}$$

This produces position-specific advantage sequences $\hat{\mathbf{A}}^j = [\hat{A}_1^j, \hat{A}_2^j, \ldots, \hat{A}_{T_j+1}^j]$ for each trajectory, where all tokens within iteration $t$ receive advantage $\hat{A}_t^j$. These advantages vary across iterations even within the same trajectory—early iterations that successfully identify critical errors may receive higher advantages than later iterations that make minor refinements. We then update the policy using these position-specific advantages with standard GRPO (Shao et al., 2024), jointly optimizing both autoformalization accuracy and self-validation quality.

## 4 EXPERIMENT

### 4.1 EXPERIMENTAL SETUP

**Datasets.** To rigorously assess the effectiveness of REFORM, we evaluate on four challenging benchmarks: **(1) miniF2F** (Zheng et al., 2021): 244 test problems from high-school mathematics competitions. **(2) ProofNet** (Azerbayev et al., 2023): 186 undergraduate-level theorems from textbooks spanning real analysis, abstract algebra, and topology. **(3) PutnamBench** (Tsoukalas et al., 2024): 644 college-level competition problems from the Putnam Mathematical Competition (1962-2023). **(4) AIME2025** (OpenCompass, 2025): 30 problems from the 2025 American Invitational Mathematics Examination, testing autoformalization on contemporary competition problems.

**Baselines.** We compare our REFORM against the state-of-the-art methods including: **(1) Proprietary and Open-source Models**: We evaluate frontier LLMs including GPT-5 (OpenAI, 2025), Claude-3.7-Sonnet (Anthropic, 2025), Gemini-2.5-Pro (Google, 2025), DeepSeek-R1-0528 (Guo et al., 2025), QwQ-32B (Qwen, 2024), and Qwen3 series (Yang et al., 2025). **(2) Autoformalization Models**: We compare with state-of-the-art autoformalization models, including DeepSeek-Prover-V1.5-RL (Xin et al., 2024b), Goedel-V1 (Lin et al., 2025a), Kimina-Autoformalizer-7B (Wang et al., 2025a), and Goedel-FormalizerV2 (8B and 32B) (Lin et al., 2025b).

**Evaluation Metrics.** We consider two key metrics: **(1) Syntactic Correctness (syn)**: whether the formal statement passes Lean compiler verification; **(2) Semantic Consistency (sem)**: whether the statement is both syntactically correct and semantically faithful. This is our primary metric. We adopt `Qwen3-235B-A22B` as our default evaluation model unless explicitly stated otherwise.

Table 1: Main results. We report both syntactic correctness (syn) and semantic consistency (sem), with sem being our primary metric. ‡Improvements are relative to the best baseline with comparable model size. The best results are in **bold**, and the second best are underlined among baselines.

| Model | miniF2F | | ProofNet | | Putnam | | AIME2025 | | AVG | |
|---|---|---|---|---|---|---|---|---|---|---|
| | syn | sem | syn | sem | syn | sem | syn | sem | syn | sem |
| *Proprietary and Open-source Models* | | | | | | | | | | |
| GPT 5 | 70.9 | 66.0 | 49.5 | 44.6 | 61.6 | 45.8 | 13.3 | 13.3 | 48.8 | 42.4 |
| Claude-3.7-sonnet | 40.2 | 34.0 | 28.5 | 22.0 | 20.2 | 10.1 | 3.3 | 3.3 | 22.2 | 16.5 |
| Gemini-2.5-pro | 28.7 | 25.8 | 23.1 | 8.1 | 15.7 | 6.5 | 13.3 | 0.0 | 20.2 | 10.1 |
| DeepSeek-R1-0528 | 38.5 | 35.2 | 11.3 | 9.7 | 19.7 | 11.3 | 16.7 | 3.3 | 21.6 | 14.9 |
| Qwen3-235B-A22B | 55.7 | 43.9 | 16.7 | 12.9 | 33.1 | 19.9 | 20.0 | 13.3 | 31.4 | 22.5 |
| Qwen3-32B | 57.4 | 53.3 | 10.8 | 8.1 | 8.2 | 6.2 | 10.0 | 10.0 | 21.6 | 19.4 |
| Qwen3-8B | 37.7 | 31.6 | 7.5 | 5.9 | 4.5 | 3.1 | 3.3 | 0.0 | 16.6 | 10.1 |
| *Autoformalization Models* | | | | | | | | | | |
| DeepSeek-Prover-V1.5-RL | 86.1 | 43.0 | 36.6 | 16.1 | 11.3 | 8.6 | 0.0 | 0.0 | 30.8 | 14.8 |
| Goedel-V1-32B-Workbook | 95.1 | 47.1 | 48.4 | 18.3 | 62.1 | 9.3 | 70.0 | 3.3 | 68.9 | 19.5 |
| Goedel-V1-32B-Sonnet | 93.4 | 69.3 | 47.8 | 26.3 | 73.1 | 17.2 | 80.0 | 13.3 | 73.6 | 31.5 |
| Kimina-Autoformalizer-7B | 92.6 | 67.6 | 53.2 | 23.6 | 69.7 | 25.1 | 80.0 | 16.7 | 73.9 | 33.3 |
| Goedel-Formalizer-V2-8B | 97.5 | 81.1 | 70.4 | 47.3 | 74.5 | 42.9 | 66.7 | 26.7 | 77.3 | 49.5 |
| Goedel-Formalizer-V2-32B | 97.1 | 82.0 | 71.5 | 50.5 | 74.2 | 41.4 | 66.7 | 26.7 | 77.3 | 50.1 |
| *Ours* | | | | | | | | | | |
| REFORM-8B | **98.4** | 87.7 | 78.5 | 65.6 | 81.9 | 57.3 | 83.3 | 46.7 | 85.5 | 64.3 |
| + Improvement‡ | ↑0.9 | ↑6.6 | ↑8.1 | ↑18.3 | ↑7.4 | ↑14.4 | ↑3.3 | ↑20.0 | ↑8.2 | ↑14.8 |
| REFORM-32B | 97.1 | **91.4** | **82.3** | **70.4** | **83.1** | **62.3** | **86.7** | **66.7** | **87.3** | **72.7** |
| + Improvement‡ | ↑0.0 | ↑9.4 | ↑10.8 | ↑19.9 | ↑8.9 | ↑20.9 | ↑6.7 | ↑40.0 | ↑10.0 | ↑22.6 |

**Implementation Details.** We implement REFORM based on Qwen3 (8B and 32B) (Yang et al., 2025). We curate training data from diverse open sources including Omni-MATH (Gao et al., 2024), IneqMath (Jiayi et al., 2025), the Lean Workbook (Ying et al., 2024), DeepTheorem (Zhang et al., 2025), Natural Proofs (Razborov & Rudich, 1994), and Big-Math (Albalak et al., 2025), with rigorous deduplication against all test sets to ensure fair evaluation. For details, we refer readers to Appendix A.1 for dataset statistics, SFT and RL training procedures.

## 4.2 MAIN RESULTS

Table 1 presents comprehensive evaluation results across four challenging autoformalization benchmarks. Additionally, we provide `CriticLean` based evaluation results to further validate the robustness of our findings (Appendix 4.6). We have two key findings that validate the effectiveness of the reflective paradigm.

**First**, our REFORM achieves state-of-the-art performance with substantial improvements across all benchmarks. REFORM-8B surpasses the strongest baseline Goedel-FormalizerV2-8B by an average of +14.8pp in semantic consistency while improving syntactic correctness (+8.2pp). Remarkably, our 8B model even outperforms the 4× larger Goedel-FormalizerV2-32B by +14.2pp in semantic consistency, demonstrating that the reflective paradigm's architectural innovation goes byond mere parameter scaling. The improvements are particularly pronounced on the more challenging benchmarks: +18.3pp on ProofNet and +14.4pp on PutnamBench compared to Goedel-V2-8B, with the most dramatic gain of +20.0pp on AIME2025. These consistent improvements across diverse benchmarks confirm that iterative self-validation fundamentally enhances semantic understanding.

**Second**, the reflective paradigm specifically addresses the critical semantic issue that plagues existing autoformalization systems. **All baseline methods exhibit a severe syntactic-semantic performance gap**, revealing that models readily generate Lean-compilable code but systematically fail at semantic fidelity. This gap is most extreme in the more difficult datasets (e.g. Putname). In contrast, REFORM's improvements are more concentrated on semantic consistency than on syntactic refinement. This asymmetric improvement pattern amplifies on harder benchmarks, with semantic gains consistently exceeding syntactic improvements by 2-3×, demonstrating that by making semantic validation an integral part of generation, REFORM transforms autoformalization from superficial pattern matching to genuine mathematical understanding.

## 4.3 ABLATION STUDIES

We conduct comprehensive ablation studies to analyze the contribution of each component in our framework, as shown in Table 2. The experiments are divided into two parts: **(1) Training Components.** Removing the bounded clipping in Eq. 3 causes severe degradation particularly on the harder benchmarks, confirming that bounding returns is crucial for stable optimization with heterogeneous rewards. The auxiliary reward $r_{aux}$ exhibits increasing importance as problem complexity increases, indicating that explicit supervision for self-validation quality be-

Table 2: Ablation studies on training methodology and paradigm with semantic consistency score. All variants use identical training data to ensure fair comparison.

| Method | miniF2F | ProofNet | Putnam | AIME25 |
|---|---|---|---|---|
| *Ablation on training Methodology* | | | | |
| REFORM | **87.7** | **65.6** | **57.3** | **46.7** |
| w/o clip | 84.0 | 59.6 | 48.9 | 26.7 |
| w/o $r_{aux}$ | **87.7** | **65.6** | 52.1 | 40.0 |
| w/o RL | 85.2 | 62.3 | 49.4 | 30.0 |
| *Ablation on Paradigm* | | | | |
| One-pass | 82.7 | 59.1 | 40.8 | 16.7 |

comes more critical for harder problems. The RL training phase provides consistent improvements across all datasets, with gains increasing on harder problems, demonstrating that PBSO effectively learns complex reasoning strategies beyond SFT. **(2) Paradigm Comparison.** The most striking result emerges from comparing our reflective paradigm against one-pass generation: when trained on identical data, the one-pass baseline shows dramatic performance gaps that widen with problem difficulty. This widening gap validates our core hypothesis: as mathematical complexity increases, the need for iterative self-correction becomes paramount. Single-pass generation fundamentally lacks the mechanism to identify and rectify its own semantic errors, while our reflective paradigm enables progressive refinement through self-validation.

## 4.4 TRAINING DYNAMICS OF PBSO

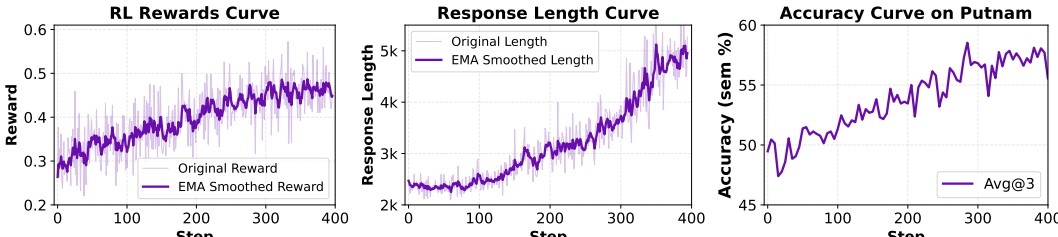

Figure 3: Training dynamics of our RL process.

To understand how Prospective Bounded Sequence Optimization enables effective learning of the reflective paradigm, we analyze the training dynamics in REFORM-8B across three key dimensions. Figure 3 reveals three critical insights into how PBSO shapes model behavior:

**(1) Stable Heterogeneous-Objective Optimization.** The training reward (left) steadily improves from 0.30 to 0.47 over 400 steps, with progressively narrowing confidence bands indicating stable convergence. This smooth progression demonstrates that PBSO successfully balances heterogeneous rewards at different sequence positions, optimizing both task success and critique quality. **(2) Emergent Reflective Behavior.** Response length (middle) exhibits remarkable organic growth from 2,300 to 4,800 tokens during training—a 2.1× expansion solely from heterogeneous reward signals, without any explicit length bonuses or penalties. This phenomenon reveals a crucial insight: when properly incentivized through auxiliary rewards for critique quality, models autonomously develop more thorough self-examination behaviors. **(3) Robust Generalization.** Performance on held-out PutnamBench (right) improves from 47% to 57% in semantic consistency, closely tracking training rewards. This tight correlation between training and test performance, maintained throughout optimization rather than diverging due to overfitting, demonstrates that PBSO enables learning of transferable reflective capabilities. Together, these dynamics reveal how PBSO orchestrates the RL process: heterogeneous rewards drive the emergence of reflective behavior, which in turn generates richer training signals, further improving both autoformalization and self-validation capabilities in a virtuous cycle.

## 4.5 RELIABILITY OF SEMANTIC CONSISTENCY EVALUATION

Since our evaluation relies on LLM-based judges to assess semantic consistency, establishing their reliability is crucial for validating our experimental conclusions. We construct **ConsistencyCheck**, a benchmark of 859 expert-annotated items where models perform binary classification: determining whether a formal statement correctly preserves the mathematical semantics of the original question.

**Human expert fallibility in existing benchmarks.** During the annotation process, we uncovered that 16.4% of miniF2F and 38.5% of ProofNet's human-written formal statements contain semantic errors. This high error rate in expert-crafted formalizations underscores that autoformalization challenges even human specialists, further motivating the need for automated approaches like REFORM.

Table 3: LLM performance on ConsistencyCheck benchmark for semantic consistency evaluation. [†]Full model names: Claude-3.7-Sonnet, Qwen3-235B-A22B-Thinking, CriticLean-14B.

| Metrics | GPT-5 | Gemini-2.5-pro | Claude-3.7[†] | DeepSeek-R1 | Qwen3-235B[†] | QwQ | CriticLean[†] |
|---|---|---|---|---|---|---|---|
| **Accuracy** | 82.5 | **85.8** | 77.2 | 78.1 | 82.9 | 77.9 | 79.1 |
| **Precision** | **88.9** | 84.4 | 75.7 | 84.7 | 85.3 | 75.5 | 80.7 |
| **Recall** | 82.9 | **96.9** | 93.3 | 79.0 | 87.7 | 95.4 | 87.3 |
| **F1** | 85.8 | **90.2** | 83.6 | 81.8 | 86.5 | 84.3 | 83.9 |

**LLM evaluation reliability analysis.** Table 3 reveals that while Gemini-2.5-Pro achieves the highest accuracy (85.8%), open-source Qwen3-235B-A22B provides comparable performance (82.9%) with balanced precision-recall trade-offs. These results reveal two critical insights for the autoformalization community: (1) **Classification-Generation Gap validates autoformalization's difficulty.** On this classification task, which is inherently simpler than generation, frontier models plateau at 86% accuracy. This 14% error rate in merely *recognizing* semantic consistency helps explain why *generating* faithful formalizations remains fundamentally challenging, as generation requires not just recognition but creative synthesis under semantic constraints. (2) **Current evaluation is sufficiently reliable for our experiments.** Despite imperfections, with an accuracy at $85.8\%$, current LLMs provide adequate signals for drawing research conclusions. Crucially, REFORM's improvements far exceed potential evaluation noise: our +14.4pp gain on PutnamBench represents a 2.5-standard-deviation effect size given the judge's error rate, while our +20.0pp improvement on AIME2025 corresponds to 3.5 standard deviations — both statistically robust. Based on these analyses, we adopt `Qwen3-235B-A22B` as our primary semantic judge (balancing quality with reproducibility) and `CriticLean-14B` for RL training (for efficiency). Finally, ConsistencyCheck is released to facilitate future research on autoformalization evaluation reliability.

**Human Evaluation on REFORM**   While LLM-based evaluation provides scalable assessment, we further conduct human evaluation to directly validate REFORM's outputs. We evaluate the final formal statements generated by REFORM-8B on miniF2F and ProofNet test sets. Each statement was classified as "Correct" only if it was both syntactically valid and semantically faithful to the original problem; otherwise, it was deemed "Incorrect". The human evaluation revealed high fidelity, with 86.1% of miniF2F and 69.4% of ProofNet formalizations verified as correct, closely aligning with our LLM-based semantic consistency scores (87.7% and 65.6% respectively) and validating the reliability of automated evaluation.

## 4.6 ROBUSTNESS ANALYSIS ON MAIN RESULT EVALUATED BY CRITICLEAN-14B

To validate that our improvements are not artifacts of a specific evaluation metric, Table 4 presents results evaluated by CriticLean-14B (Peng et al., 2025), an independent semantic consistency judge trained specifically for autoformalization assessment.

**(1) Consistent Improvements Across Evaluators.** REFORM maintains substantial advantages under CriticLean evaluation, with ReForm-8B achieving an average improvement of +18.0pp over Goedel-V2-8B (compared to +14.8pp under Qwen3-235B evaluation). This consistency across fundamentally different evaluators—a general-purpose LLM (Qwen3) versus a specialized critic model (CriticLean)—strongly validates the robustness of our approach.

**(2) Amplified Gains on Challenging Benchmarks.** The improvements are even more pronounced under CriticLean evaluation for difficult datasets: +33.3pp on AIME2025 (vs +20.0pp with Qwen3)

Table 4: Main results. We report semantic consistency (sem) based on `CriticLean-14B` (Peng et al., 2025) with sem being our primary metric. [‡]Improvements are relative to the best baseline with comparable model size. The best results are in **bold**, and the second best are underlined.

| Model | miniF2F | ProofNet | Putnam | AIME2025 | AVG |
|---|---|---|---|---|---|
| *Proprietary and Open-source Models* | | | | | |
| GPT-5 | 66.8 | 41.5 | 44.1 | 13.3 | 41.7 |
| Claude-3.7-Sonnet | 34.8 | 22.6 | 10.9 | 0.0 | 17.1 |
| Gemini-2.5-Pro | 28.3 | 7.0 | 4.5 | 0.0 | 10.0 |
| DeepSeek-R1-0528 | 33.2 | 10.2 | 11.2 | 3.3 | 14.5 |
| Qwen3-235B-A22B | 44.7 | 12.4 | 18.8 | 23.3 | 24.8 |
| Qwen3-32B | 54.1 | 7.5 | 6.7 | 6.7 | 18.8 |
| Qwen3-8B | 32.0 | 7.0 | 3.1 | 0.0 | 10.5 |
| *Autoformalization Models* | | | | | |
| DeepSeek-Prover-V1.5-RL | 44.3 | 0.0 | 0.5 | 0.0 | 11.2 |
| Goedel-V1-32B-Workbook | 48.8 | 18.3 | 9.6 | 3.3 | 20.0 |
| Goedel-V1-32B-Sonnet | 66.0 | 23.1 | 17.4 | 10.0 | 29.1 |
| Kimina-Autoformaiizer-7B | 66.8 | 22.0 | 26.2 | 13.3 | 32.1 |
| Goedel-Formalizer-V2-8B | 86.9 | 54.8 | 40.8 | 26.7 | 52.3 |
| Goedel-Formalizer-V2-32B | 89.3 | 59.1 | 44.3 | 33.3 | 56.5 |
| *Ours* | | | | | |
| REFORM-8B | **92.2** | 69.4 | 59.6 | 60.0 | 70.3 |
| + Improvement[‡] | ↑ 5.3 | ↑ 14.6 | ↑ 18.8 | ↑ 33.3 | ↑ 18.0 |
| REFORM-32B | 91.4 | **73.7** | **64.6** | **63.3** | **73.2** |
| + Improvement[‡] | ↑ 2.1 | ↑ 14.6 | ↑ 20.3 | ↑ 30.0 | ↑ 16.7 |

and +18.8pp on PutnamBench (vs +14.4pp). This suggests that CriticLean may be more sensitive to semantic nuances in complex problems, where our reflective paradigm provides the greatest benefits.

**(3) Different Absolute Scores but Consistent Rankings.** While CriticLean generally assigns higher semantic consistency scores than Qwen3 (possibly due to different training objectives or calibration), the relative rankings remain largely consistent. Notably, REFORM achieves the highest scores under both evaluators, with ReForm-8B reaching 70.3% average semantic consistency under CriticLean—a remarkable achievement given the inherent difficulty of autoformalization.

These results from Table 1 and Table 4 confirm that the reflective paradigm's effectiveness transcends specific evaluation methodologies, providing robust improvements in semantic consistency regardless of how it is measured.

## 5 CONCLUSION

We introduce REFORM, a reflective autoformalization paradigm that fundamentally shifts from one-pass generation to an iterative process interweaving generation with semantic self-validation. To effectively train the reflective paradigm, we propose Prospective Bounded Sequence Optimization, which realizes heterogeneous rewards via prospective bounded returns. This enable models to jointly optimize for both correct final formalizations and accurate intermediate critiques, preventing degenerate or hallucinated self-validations. Extensive experiments demonstrate the effectiveness of our REFORM with an average improvement of 22.6% across four benchmarks. Our ConsistencyCheck benchmark further reveals that autoformalization challenges even human experts while confirming the reliability of LLM-based evaluation metrics.

ACKNOWLEDGMENTS

This work was supported by Alibaba Research Intern Program. This paper was partially supported by the National Natural Science Foundation of China No. 92470205 and Beijing Major Science and Technology Project under Contract No. Z251100008425002.

ETHICS STATEMENT

Our work focuses on advancing automated mathematical formalization through a novel reflective paradigm and reinforcement learning algorithm. We have carefully considered the ethical implications of our research and taken appropriate measures to ensure responsible development.

**Human Annotation and Labor.** The construction of our ConsistencyCheck benchmark involved human experts in mathematics and Lean4. All annotators were fairly compensated for their expertise and time according to prevailing standards for skilled technical work. The annotation process was designed to be intellectually engaging rather than repetitive, leveraging the annotators' mathematical expertise. We ensured reasonable working conditions with no excessive time pressures, and annotators retained the right to decline or withdraw from tasks at any point.

**Intellectual Property and Attribution.** Our benchmark builds upon existing mathematical datasets (miniF2F and ProofNet), which we use in accordance with their licenses and with proper attribution. We acknowledge the substantial human effort that went into creating these original resources and ensure all sources are appropriately cited.

**Potential Impacts.** While our work aims to democratize access to formal mathematical reasoning tools, we acknowledge potential concerns. The automation of mathematical formalization could reduce demand for certain types of mathematical verification work. However, we believe our technology will primarily augment rather than replace human mathematicians, enabling them to focus on higher-level creative and conceptual work. Furthermore, by making formal verification more accessible, our work could enhance mathematical education and research, particularly in resource-constrained settings.

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

# A    MORE IMPLEMENTATION DETAILS

In this section, we provide a comprehensive implementation details of our proposed method. For additional insights and more intricate details, we refer the reader to our supplementary materials.

## A.1    IMPLEMENTATION DETAILS

Table 5: Key hyperparameters in the SFT phase.

| Hyperparameter | Value |
|---|---|
| Learning Rate | 1e-5 |
| Batch size | 512 |
| #Epochs | 3 |
| Chat template | Qwen |
| Max Context Length | 40960 |
| Warmup ratio | 0.03 |
| LR scheduler type | Cosine |

Table 6: Key hyperparameters in the RL phase.

| Hyperparameter | Value |
|---|---|
| Learning Rate | 1e-6 |
| Batch size | 32 |
| Group size per Question ($G$) | 16 |
| Temperature | 1.0 |
| Top-p | 0.95 |
| KL coefficient ($\lambda$) | 0.0 |
| Entropy coefficient | 0.0 |

**Supervised Fine-turning Phase.**    We utilize `Slime`[2] as our training framework for the initial supervised fine-tuning phase. The detailed hyper-parameters for this phase are presented in Table 5. Since current LLMs lack inherent capabilities for our iterative autoformalization paradigm, we first construct high-quality training data that demonstrates both autoformalization and semantic self-validation behaviors. We employ a multi-agent system based on `Qwen3-235B-A22B-Thinking` to generate training trajectories that embody our reflective paradigm. For each mathematical problem in our source datasets (Section 4.1), we execute the following iterative process:

- (Step 1) **Initial Autoformalization:** Generate an initial Lean4 formalization using a standard one-pass prompt (Appendix D.1).
- (Step 2) **Syntactic Validation:** Verify the generated statement compiles successfully in Lean4. Non-compiling statements trigger subsequent refinement.
- (Step 3) **Semantic Consistency Evaluation:** For syntactically valid statements, apply a consistency checking prompt (Appendix D.2) to assess semantic alignment with the original problem.
- (Step 4) **Reflective Refinement:** When inconsistencies are detected, combine the failed statement with evaluation feedback to generate corrections using reflective prompts (Appendix D.1, D.2).
- (Step 5) **Iteration:** Repeat steps 2-4 until either (a) semantic consistency is achieved, (b) maximum iterations (3 rounds) are reached, or (c) no further improvements are generated.

This pipeline produces training trajectories that naturally interweave autoformalization attempts with self-validation and correction, providing rich supervision for learning our reflective paradigm. The resulting dataset contains 447,508 trajectories with iteration distributions detailed in Table 7, where 83.1% of problems achieve resolution within a single iteration while the remaining require multiple rounds of refinement.

Table 7: Data Statistics for SFT and RL Phase.

| Stage | Data Type | Iteration Distribution | | | Total |
|---|---|---|---|---|---|
| | | Iteration 1 | Iteration 2 | Iteration 3 | |
| SFT | Trajectories | 371,679 | 65,734 | 10,095 | 447,508 |
| RL | Question only | - | - | - | 2,048 |

**Reinforcement Learning Phase.**    For training data in RL phase, we start with a diverse batch of mathematical problems spanning various difficulties, grade levels, and domains. For each problem, we generate 8 candidate formalizations by sampling from our SFT model. These candidates were then evaluated against two successive criteria: compiler verification and a semantic consistency

---

[2]https://github.com/THUDM/slime

check. From this pool, we curate a final dataset with 2048 items for RL. The selection was deliberately stratified to include problems with varying pass rates (i.e., the proportion of the eight samples that passed the checks), thereby ensuring the dataset represented a wide spectrum of formalization difficulty. Moreover, Table 6 summarizes the key hyperparameters used during the reinforcement learning phase. We also use Slime as our RL frameowrk due to its efficient and easy to use.

**Inference Phase.** During inference, we employ deterministic sampling with temperature 0.6 and top-p 0.95 to balance between generation quality and diversity. The maximum generation length is set to 40,960 tokens. We utilize vLLM[3] as our inference engine.

## A.2 IMPACT OF RL TRAINING ON ITERATIVE REFINEMENT BEHAVIOR

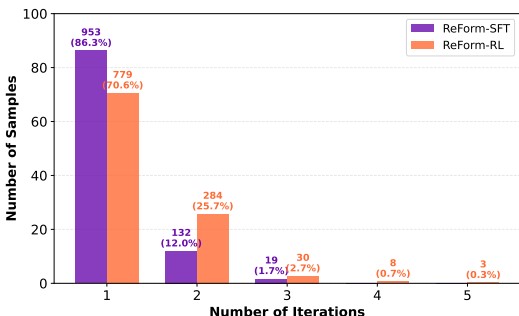

Figure 4: Iteration Distribution of our REFORM-SFT and RL.

We further analyze the distribution of iteration rounds for ReForm-8B across four benchmarks, comparing models after SFT and RL training stages. As shown in Figure 4, the results reveal striking behavioral changes that validate our reflective paradigm's effectiveness.

REFORM-SFT exhibits a heavily skewed distribution, with 86.3% of samples terminating after a single iteration and the remaining samples distributed across 2-3 iterations—a pattern directly reflecting our SFT training data, which was capped at three iterations. In contrast, REFORM-RL demonstrates a markedly different behavioral pattern. Most notably, it explores iteration depths never seen during SFT training, with 0.7% and 0.3% of samples extending to 4 and 5 iterations respectively. More importantly, the overall distribution shifts toward deeper refinement: the percentage of single-iteration completions drops to 70.6%, while 2-iteration cases nearly double from 12.0% to 25.7%. This redistribution indicates that **PBSO training successfully teaches the model to recognize when additional refinement is beneficial, rather than prematurely terminating the reflective process**. The emergence of 4-5 iteration trajectories—patterns entirely absent from the SFT training data—demonstrates that PBSO enables genuine exploration beyond the supervised distribution, discovering more effective refinement strategies through trial and error. This finding is further corroborated by the consistent increase in average response length during RL training (Section 4.4), confirming that the model learns to invest more computational effort in challenging problems that benefit from extended reflection.

The results provide strong empirical evidence that our heterogeneous reward mechanism successfully prevents the degeneration of self-validation capabilities while encouraging productive iteration when needed, ultimately validating the core premise of our reflective autoformalization paradigm: that iterative self-correction, when properly incentivized, leads to superior semantic consistency.

## B THE CONSISTENCYCHECK BENCHMARK

**Annotation Team.** Our annotation team for both the ConsistencyCheck benchmark and the autoformalization results in Section 4.5 comprised 6 members. All are senior PhD. candidates with a strong background in mathematical competitions and prior experience in formalization-related annotation tasks.

---

[3] https://github.com/vllm-project/vllm

**Expertise & Training.** To ensure high-quality and consistent annotations, a rigorous training protocol was implemented. This included dedicated sessions on interpreting the annotation guidelines, mastering the criteria for semantic consistency, and standardizing the handling of ambiguous or edge cases. Furthermore, all annotators are active researchers in Lean and formalization-related fields, possessing practical experience in formal proof development or autoformalization.

**Annotation Protocol.** Our annotation protocol employed a three-annotator design per statement to ensure robustness. Initially, two annotators worked independently. Subsequently, a third senior annotator reviewed their annotations and accompanying textual comments (examples of which are provided in Appendix C) to perform cross-validation, resolve any discrepancies, and render the final judgment.

**Annotation process.** To construct a high-quality benchmark for evaluating semantic consistency, we commission a team of experts with deep proficiency in both mathematics and Lean4. The annotators were tasked with assessing the semantic fidelity of formal statements from a dataset composed of items from miniF2F (Zheng et al., 2022) and ProofNet (Azerbayev et al., 2023), which were presented to them in an anonymized format. For each item, **two experts** independently compared the machine-verifiable formal statement against the original natural language problem. If the formalization faithfully captured the problem's semantic intent, it was labeled as "Correct". If any semantic discrepancy was found, it was labeled as "Incorrect", and the annotators were required to provide a detailed written justification. In cases of disagreement, **a third senior expert** was brought in to adjudicate and determine the final label, ensuring the reliability of our benchmark.

**Example of our ConsistencyCheck Benchmark.** A representative example of a semantic error discovered in a ProofNet entry is illustrated in Example B. In this instance, the human-authored formalization contained two critical flaws: (1) a "constant term mismatch", where $\sqrt{11}$ from the natural language was incorrectly transcribed as $11$; and (2) a "degree bound inconsistency", where the constraint "degree $\leq 80$" was altered to "degree $< 80$". Notably, the annotation team also identified another significant class of error in miniF2F entries: cases where the formal statement included an explicit answer, while the original natural language problem did not. These instances were also systematically classified as "Incorrect." Such discrepancies underscore the profound difficulty of achieving true semantic fidelity, even in expert-curated datasets.

Based on this benchmark, we investigate the performance of each LLMs in semantic consistency evaluation. These models are tasked with assessing the semantic consistency following the prompt specified in Appendix D.2, as discussed in Section 4.5.

---

**Examples in ConsistencyCheck Benchmark**

**Example 1**

**set:** ProofNet
**name:** exercise_5_4_3
**split:** test
**question:** If $a \in \mathbb{C}$ is such that $p(a) = 0$, where $p(x) = x^5 + \sqrt{2}x^3 + \sqrt{5}x^2 + \sqrt{7}x + \sqrt{11}$, show that $a$ is algebraic over $\mathbb{Q}$ of degree at most 80.
**header:**

```
import Mathlib
open Fintype Set Real Ideal Polynomial
open scoped BigOperators
```

**formal statement:**

```
theorem exercise\_5\_4\_3 {a : ℝ} {p : ℝ → ℝ} \n  (hp : p =
    λ x, x\^{}5 + real.sqrt 2 * x\^{}3 + real.sqrt 5 * x\^{}2
    + \n  real.sqrt 7 * x + 11)\n  (ha : p a = 0) : \n  ∃ p
    : polynomial ℝ, p.degree < 80 ∧ a ∈ p.roots ∧ \n  ∀ n : p
    .support, ∃ a b : ℤ, p.coeff n = a / b :=
```

**Human check**
Incorrect
**Human comments**
1. Polynomial constant term mismatch: $\sqrt{11}$ in natural language vs 11 in formalization.
2. Degree bound inconsistency: degree $\leq 80$ in natural language vs $< 80$ (i.e. $\leq 79$) in formalization.
These discrepancies make the formalization mathematically inequivalent to the original statement.

| **Example 2** |
| --- |

**set:** miniF2F
**name:** aime_1991_p6
**split:** test
**question:** Suppose $r$ is a real number for which

$$\left\lfloor r + \frac{19}{100} \right\rfloor + \left\lfloor r + \frac{20}{100} \right\rfloor + \left\lfloor r + \frac{21}{100} \right\rfloor + \cdots + \left\lfloor r + \frac{91}{100} \right\rfloor = 546.$$

Find $\lfloor 100r \rfloor$. (For real $x$, $\lfloor x \rfloor$ is the greatest integer less than or equal to $x$.) Show that it is 743.
**header:**

```
import Mathlib\nimport Aesop\n\nset_option maxHeartbeats 0\
    nopen BigOperators Real Nat Topology Rat\n\n
```

**formal statement:**

```
theorem aime_1991_p6 (r : ℝ) (h₀ : (∑ k in Finset.Icc (19 :
    ℕ) 91, Int.floor (r + k / 100)) = 546) :
  Int.floor (100 * r) = 743 := by
```

**Human check**
Correct

## C   CASE STUDY OF OUR REFORM

We present a detailed case study demonstrating our model's reflective autoformalization capabilities in practice. The example below illustrates how REFORM successfully identifies and corrects its own semantic errors through iterative self-validation.

**Round 1: Initial Attempt and Error Detection.** The model generates its first autoformalization attempt, then performs semantic self-validation by systematically analyzing the formalization across five critical dimensions: (1) variable definitions and types, (2) mathematical constraints and conditions, (3) logical relationships between statements, (4) completeness of the formalization, and (5) alignment with the problem's intent. Through this structured analysis, the model autonomously identifies semantic inconsistencies in its initial output (highlighted in red).

**Round 2: Self-Correction and Validation.** Leveraging its self-generated critique as feedback, the model produces a refined autoformalization that directly addresses the identified issues. The corrected statement undergoes another round of semantic validation, this time passing all consistency checks. This successful self-correction demonstrates the model's ability to not only detect its own errors but also synthesize this understanding into improved formalizations.

This example showcases the core strength of our reflective paradigm: the model's capacity for genuine self-improvement without external intervention, achieving semantic correctness through iterative refinement guided by its own critical analysis.

---

**Example of Our REFORM**

**Question:** What value of $x$ will give the minimum value of $x^2 - 14x + 3$? Show that it is 7.

**The response of our REFORM:**
<**think**>

**Round 1 (Autoformalization)**

<**round**>
The problem is asking for the value of $x$ that minimizes the expression $x^2 - 14x + 3$, and it states that this value is 7.
To formalize this in Lean 4, I need to express that: 1. The function $f(x) = x^2 - 14x + 3$ has a minimum at $x = 7$ 2. This means that for all real numbers $x$, we have $f(7) \leq f(x)$
Let me think about how to express this properly in Lean 4:

```lean4
theorem test_problem : ∀ x : ℝ, (7 : ℝ)^2 - 14 * 7 + 3 ≤ x^2
    - 14 * x + 3 := by sorry
```

Wait, let me think about this more carefully. The problem is asking "what value of x will give the minimum value". This suggests we want to find the argmin of the function.
Actually, let me be more precise. The statement should be that 7 is the point where the function achieves its minimum. So we want:

```lean4
theorem test_problem : ∀ x : ℝ, (7 : ℝ)^2 - 14 * 7 + 3 ≤ x^2
    - 14 * x + 3 := by  sorry
```

But this is a bit awkward with the explicit casting. Let me try:

```lean4
theorem test_problem : ∀ x : ℝ, 7^2 - 14 * 7 + 3 ≤ x^2 - 14
    * x + 3 := by  sorry
```

Actually, in Lean 4, when we write

`7^2` where `7 : ℝ`

it should automatically be interpreted as a real number. But to be safe, let me be explicit:

```lean4
theorem test_problem : ∀ x : ℝ, (7 : ℝ)^2 - 14 * (7 : ℝ) + 3
    ≤ x^2 - 14 * x + 3 := by  sorry
```

I'll verify the calculation to ensure accuracy. The key is to confirm that 7 truly minimizes the quadratic expression by checking the algebraic properties of the function.

Expanding the squared term reveals the minimum occurs precisely at 7, with a value of -46. The algebraic manipulation shows $x^2 - 14x + 3$ reaches its lowest point when $x = 7$, supporting the original claim about the minimum value.

The Lean theorem captures this relationship, demonstrating that for all real numbers, the function's value at 7 is less than or equal to the function's value at any other point.Looking at this problem, I need to formalize the statement that x = 7 gives the minimum value for the expression x² - 14x + 3.

```lean4
import Mathlib
import Aesop

set\_option maxHeartbeats 0

open BigOperators Real Nat Topology Rat

theorem test_problem : ∀ x : ℝ, (7 : ℝ)^2 - 14 * 7 + 3 ≤ x^2
    - 14 * x + 3 := by  sorry
```

**Round 1 (Semantic Self-validation)**

It's time to assess if the Formal Statement correctly captures the essence of the given math problem.

1. Math Assertion Analysis: The problem asks for the value of $x$ that minimizes the quadratic function $x^2 - 14x + 3$ and requires proving that this value is 7. The key components are: (a) the function $f(x) = x^2 - 14x + 3$, (b) the variable $x \in \mathbb{R}$, (c) the claim that $x = 7$ is the minimizer, and (d) the requirement to prove this claim.

2. Lean Statement Analysis: The theorem asserts $\forall x : \mathbb{R}, (7^2 - 14 \cdot 7 + 3) \leq x^2 - 14x + 3$. This captures the inequality that the function's value at $x = 7$ is less than or equal to its value at any other $x \in \mathbb{R}$. However, it does not explicitly state that $x = 7$ is the minimizer (i.e., the argument where the minimum occurs). The theorem only asserts the inequality, not the existence of the minimizer or its uniqueness.

3. Comparative Verification: The Lean statement correctly captures the inequality part of the problem (that f(7) is the minimum value). However, it does not explicitly state that x=7 is the value that achieves this minimum. The problem requires both identifying the minimizer (x=7) and proving the inequality. The Lean theorem omits the explicit identification of x=7 as the minimizer, focusing only on the inequality.

4. Conclusion: The Lean statement is incomplete as a formalization of the problem. While it correctly captures the inequality that f(7) is the minimum value, it does not explicitly state that x=7 is the value that achieves this minimum. The problem requires both the identification of the minimizer and the proof of the inequality, but the theorem only addresses the latter.

5. Accuracy Confirmation: The Lean statement is incorrect. It fails to explicitly state that x=7 is the minimizer, which is a critical part of the problem. The theorem only asserts the inequality, not the identification of the minimizer.

Therefore, the outcome of the consistency verification is:
Incorrect

**</round>**

**Round 2 (Autoformalization)**

Time to reformulate the Lean 4 Formal Statement based on verification analysis.

**<round>**

## Thought Process

Looking at the informal statement and the critic, I see the issue with the previous formalization:

1. The problem asks for two things: - What value of x minimizes x² - 14x + 3 - Show that this value is 7

2. The previous formalization only captured the second part (that 7 gives the minimum value) but didn't explicitly state that 7 is the value that minimizes the expression.

3. To properly formalize this, I need to: - Define the function f(x) = x² - 14x + 3 - State that 7 is the unique minimizer of this function

4. In Lean, a proper formalization would need to express both that 7 is a minimizer and that it's the unique minimizer.

Given New Formal Statement:

```lean4
import Mathlib.Analysis.Calculus.Deriv.Basic
import Mathlib.Data.Real.Basic

theorem test_problem :
  let f : ℝ → ℝ := λ x => x^2 - 14*x + 3;
  (∀ x : ℝ, f 7 ≤ f x) ∧ (∀ x : ℝ, f x = f 7 → x = 7) := by
    sorry
```

**Round 2 (Semantic Self-validation)**

Now I'll check if the Formal Statement completely and correctly formalizes the mathematical question.

1. Math Assertion Analysis: The problem asks for the value of $x$ that minimizes the quadratic function $f(x) = x^2 - 14x + 3$ and requires proving that this value is 7. The mathematical structure involves: (a) defining the function $f : \mathbb{R} \to \mathbb{R}$, (b) asserting that $x = 7$ is the global minimum (i.e., $\forall x, f(7) \le f(x)$), and (c) ensuring uniqueness (i.e., if $f(x) = f(7)$, then $x = 7$). The problem implicitly relies on the function's convexity (since the coefficient of $x^2$ is positive), guaranteeing a unique minimum.

2. Lean Statement Analysis: The theorem defines $f : \mathbb{R} \to \mathbb{R}$ as $x^2 - 14x + 3$. The conclusion asserts two claims: (a) $\forall x, f(7) \le f(x)$ (global minimum), and (b) $\forall x, f(x) = f(7) \to x = 7$ (uniqueness). The function definition, domain ($\mathbb{R}$), and logical structure match the problem's requirements.

3. Comparative Verification: The Lean statement correctly captures both the existence and uniqueness of the minimum at $x = 7$. The function definition matches the mathematical text. The universal quantifiers ($\forall x$) and implications ($\to$) align with the problem's logical structure. The use of $\mathbb{R}$ ensures the domain is preserved. No extraneous conditions or missing components are present.

4. Conclusion: The Lean statement accurately formalizes the problem's requirements.

5. Accuracy Confirmation: The Lean statement correctly formalizes the problem. The function definition, domain, and logical claims (global minimum and uniqueness) align perfectly with the mathematical text.

Consequently, the outcome of the consistency check is:

Correct

**</round>**

**</think>**

| Final Statement |
|---|

With successful verification, here's the final Lean 4 Formal Statement:

```lean4
import Mathlib.Analysis.Calculus.Deriv.Basic
import Mathlib.Data.Real.Basic

theorem test_problem :
  let f : ℝ → ℝ := λ x => x^2 - 14*x + 3;
  (∀ x : ℝ, f 7 ≤ f x) ∧ (∀ x : ℝ, f x = f 7 → x = 7) := by
    sorry
```

## D  PROMPT

### D.1  INSTRUCTION FOR COLLECTING SFT TRAJECTORIES

The autoformalization prompt firstly initializes the reflective data construction process by converting a natural language mathematical problem into a Lean 4 formal statement.

When an initial formalization fails the semantic consistency check, this prompt takes as input: (1) the original mathematical problem, (2) the failed formal statement, and (3) the detailed feedback from the consistency evaluation explaining the semantic discrepancies. By explicitly conditioning on both the failure case and its diagnosis, the prompt guides the model to generate targeted corrections that directly address the identified issues rather than producing entirely new attempts that might introduce different errors.

This prompt operationalizes our core insight that models can learn from their own mistakes through structured self-reflection.

---

Reflective Autoformalization for Collecting SFT trajectories

You are an expert mathematician and Lean 4 programmer. Your task is to translate the given mathematical problem in natural language into formal statement in Lean4 syntax, strictly following the guidelines below.

Guidelines
- If the previous iteration of autoformation and consistency check results are provided, analysis it and revise the autoformation according to it.
- If the math problem is algebra question, the answer will be provided as well.
- For non-standard mathematical problems, first reformulate them into standard mathematical format.
- For complex problems containing multiple independent parts, break them down into separate theorems.
- Must strictly follow Lean 4 syntax and utilize standard Lean 4 mathematical library Mathlib4 components when possible.
1. Use correct type declarations and notation conventions
2. Include necessary imports
3. Leverage existing mathlib4 definitions and theorems
4. Follow proper naming conventions

---

- Pay attention to the consistency between the Natural Language Statement and the Formal Statement in Lean4:
1. Variable domains (e.g., $\mathbb{N}$, $\mathbb{Z}$, $\mathbb{R}$, $\mathbb{R}_+$)
2. Boundary conditions (especially for special values like 0,1)
3. Quantifier scopes ($\forall$, $\exists$)
4. Prerequisites and assumptions
5. Logical implications ($\rightarrow$, $\leftrightarrow$, $\wedge$, $\vee$)
6. Function types and properties
7. Set-theoretic notations
- Only generate the translation. Do not try to solve or prove the problem.
- Include clear documentation comments for theorems.

Input Format
Informal Statement: [Natural language description of math problem]
History of Formal Statement and consistency Comments: [The existing Lean 4 formalization and its critique, or an empty string if none exists.]

Thought Process
- Formatted Mathematical Problem
1. List all known conditions
2. Define variables and their domains
3. State assumption
- For multiple sub-problems:
Sub-problem 1: [Description]
Sub-problem 2: [Description]
...
- For revised autofomalization
1. Analyze ambiguities with natural language and potential mismatches
2. Evaluate consistency comments' validity
3. Identify missing assumptions or incorrect type signatures
4. Determine required mathlib imports in Lean 4
5. Preserve original theorem name unless invalid

Output Format
Given New Formal Statement should always use ```lean4 to start the code block and ``` to end it:
```lean4
[Corrected Lean4 code]
```

Now! It's your turn to generate the Formal Statement.
Informal Statement: {INFORMAL STATEMENT (QUESTION) HERE.}
History Formal Statement and critics: {HISTORY HERE.}
Thought:

The model's output here.

## D.2 INSTRUCTION FOR SEMANTIC CONSISTENCY CHECK

This section presents the semantic consistency evaluation prompt that serves as the foundation for both our training and evaluation procedures. Specifically, this prompt template:

- Powers the `IsConsistent` reward function during RL training, providing binary semantic correctness signals
- Drives the consistency evaluation in our ConsistencyCheck benchmark, ensuring uniform assessment criteria
- A crucial component of SFT Trajectories by validating the outputs from both the initial and reflective autoformalization attempts.

## Instruction for Consistency Check

Your role is a Lean4 expert, please help me check consistency between natural language expression and its Lean4 formal statement.

### Guidelines for Consistency Check
1. Core Checking Requirements:
- When a critique from a previous autoformalization and consistency check result is provided, you must first analyze its findings and then assess their problems.
- Must carefully compare the Natural Language Statement and the Formal Statement in Lean4 through a rigorous and explicit process.
- Determine if the Lean theorem statement is an exact and faithful formalization of the mathematical problem
- If any result is Incorrect of consistency, briefly list all inconsistencies and reasons leading to the Incorrect determination in comments

### Evaluation Stages
1. Math Assertion Analysis
Identify all structurally and semantically relevant components of the mathematical problem, including variables, types, quantifiers, constraints, logic structure, conclusion, and so on. The analysis should be based on the actual content of the text.

2. Lean Statement Analysis
Extract all structurally and semantically relevant components from the Lean statement, including
- Variable domains (e.g., real numbers vs positive real numbers)
- Boundary conditions (especially for 0,1)
- Quantifier scopes
- Prerequisites and assumptions
- Logical implications

3. Comparative Verification
Check for exact correspondence between the math and Lean statements; you may refer to aspects like:
- Semantic alignment, logic structure, and quantifier correctness.
- Preservation of constraints and boundary assumptions.
- Accurate typing and use of variables.
- Syntactic validity and proper Lean usage (free from errors).
- Use of symbols and constructs without semantic drift.
- No missing elements, no unjustified additions, and no automatic corrections or completions.

4. Final Judgement
Based solely on the above analysis, judge whether the Lean statement is a correct and exact formalization of the mathematical problem.
- When a critique from a previous consistency check is provided, you must first analyze its findings and then assess their correctness. - Result must be strictly "Correct" or "Incorrect"
- Use "Correct" ONLY when 100% mathematical equivalence is confirmed

5. Accuracy Confirmation
If correct: clearly confirm why all elements match.
If incorrect: list all mismatches and explain how each one affects correctness.

### Intput Format
The Natural Language Statement:
[A math problem in Natural language]
The Formal Statement in Lean4:
```lean4

[A Lean 4 theorem statement formalizing the problem]
```
Previous round of autoformalization and sematic validation if provided:
[The existing critique, or an empty string if none exists.]

Output Format
Return exactly one XML object
<comments>
Your brief analysis:
1. Math Assertion Analysis: [...]
2. Lean Statement Analysis (Proof Ignored): [...]
3. Comparative Verification: [...]
4. Conclusion: [...]
5. Accuracy Confirmation: [...match confirmation or list of discrepancies...]
</comments>
<consistency> Correct/Incorrect</consistency>

Now! It's your turn to compare the natural language statement with the formal statement in Lean4:
The Natural Language Statement  {INFORMAL STATEMENT (QUESTION) HERE.}

The Formal Statement in Lean4:
```lean4
{FORMAL STATEMENT HERE.}
```

Previous autoformalization and consistency Comments:
{HISTORY CRITIC HERE}

Think about the consistent result:

The model's output here.

## D.3   INSTRUCTION FOR ISFAITHFULCRITIQUE IN RL

**Instruction for IsFaithfulCritique**

Your role is a Lean4 expert, helping me review the previous consistency checking results.

Guidelines for IsFaithfulCritique Check
- Thoroughly examine the previous consistency result (Correct/Incorrect) and comments. - Carefully evaluate whether the comparison results between informal statement and formal statement in previous consistency comments are correct, and if the reasoning is sufficient. - Try to identify any errors in previous consistency comments. - Try to compare the mathematical problem and the Lean4 formal statement to see if you would reach the same consistency conclusion. - If you agree with the previous consistency result, provide your consistency result as "Correct"; if you disagree, provide "Incorrect" and explain your reasons for disagreement in the comments.
Intput Format
The Natural Language Statement:
[A math problem in Natural language]
Previous Validation result:
[Previous Lean4 formal statement and semantic validation result]

Output Format
Return exactly one xml object

> \<comments\>
> Brief analysis of my semantic validation result, with improvements if needed. If previous result is not faithfull, explain why.
> \</comments\>
> \<consistency\>Correct/Incorrect\</consistency\>
>
> Now review my consistency checking result:
> The Natural Language Statement:
> {INFORMAL STATEMENT (QUESTION) HERE.}
>
> History formal statement and consistency check result:
> {PREVIOUS CONSISTENCY CHECKING COMMENTS HERE}
> {The model's output here.}

### D.4 INSTRUCTION FOR OUR REFORM

After collecting multi-turn autoformalization trajectories through our multi-agent system (Appendix A.1), we restructure these trajectories into a unified format for SFT. The key insight is to present the entire reflective process—including initial attempts, validation results, and iterative refinements—as a single model response. This allows us to train models to internalize the complete reflective paradigm within their generation process.

> **Instruction for our Reform**
>
> Think step by step to translate the mathematical problem in natural language to Lean 4, and verify the consistency.
> {informal_statement}
>
> The model's output here.

