# OpenReview forum: "ReForm: Reflective Autoformalization with Prospective Bounded Sequence Optimization"
_ICLR.cc/2026/Conference — ICLR 2026 Poster_

### Official Review · Reviewer_63fq · 2025-10-30

**Soundness:** 3
**Presentation:** 3
**Contribution:** 3
**Rating:** 4
**Confidence:** 5

**Summary:**

The paper introduces ReForm, a novel Reflective Autoformalization framework designed to convert natural language mathematics into machine-verifiable formal statements more accurately. Unlike previous one-pass translation approaches that often lose semantic fidelity, ReForm integrates semantic self-validation directly into the autoformalization process.

**Strengths:**

- The paper provides a **ConsistencyCheck benchmark** that rigorously evaluates the reliability of **LLM-based judges** and quantifies the challenges of **autoformalization**.
- Introduces **iterative reflection techniques from reasoning models** into the process of **autoformalization**.
- The paper is **well-written**.

**Weaknesses:**

- LLM is not a perfect supervisory signal, and I would like to know to what extent the capability of the judge model affects the stability of training.
- How did you handle **semantic alignment** in your method — was it **integrated into the reward design**?
- Please provide an **ablation study comparing process-level supervision and outcome-level supervision**.

**Questions:**

Please check the **Weaknesses** section.

---

> ### Author Response · Authors · 2025-11-16
>
> We thank Reviewer 63fq for their time and insightful feedback. We appreciate the positive comments on our ConsistencyCheck benchmark, the introduction of iterative reflection, and the paper's writing.
>
> We are grateful for the opportunity to clarify the raised weaknesses. We address each point below.
>
> > Response to W1:
>
> We thank the reviewer for this critical question, which perfectly aligns with one of our paper's core investigations. We wholeheartedly agree that "LLM is not a perfect supervisory signal." This is precisely why we dedicated a significant portion of our work to quantifying this imperfection, rather than ignoring it.
>
> As detailed in our General Response, we believe our transparent analysis of the judge's limitations (the 17% error rate we found) **is a primary scientific finding of our paper, not a methodological flaw**.
>
> Regarding the specific question on how the judge's capability affects training stability:
>
> 1. Different LLM as Judges Exhibit Comparable Capability
>
> While we could not feasibly retrain our entire framework with multiple different reward models during the rebuttal period, we did prospectively analyze this exact issue. In our early explorations, we experimented with different judges, including **CriticLean, Qwen3, and DeepSeek-R1**, and found their supervisory performance to be largely comparable.
>
> This empirical finding is **quantitatively supported by our ConsistencyCheck benchmark (Table 3)**, which was designed precisely to evaluate the capability of various SOTA models. As shown, the leading models all operate within a similar capability range (e.g., Gemini-2.5-Pro at 85.8% Acc, Qwen3-235B at 82.9% Acc, and CriticLean-14B at 79.1% Acc).
>
> This leads to a crucial insight: as long as a reasonably capable SOTA judge is selected (guided by our Table 3), **the paradigm of its use is far more important than minor differences between judges**. This is exactly why our work's focus is proposing ReForm, a new reflective paradigm, rather than simply treating autoformalization as a one-pass translation task as in prior work.
>
> 2. A Contribution, Not a Weakness
>
> We respectfully argue that this analysis is a **contribution** that strengthens our paper. **Instead of just picking one judge, we provide a comparative analysis (Table 3) that helps the entire community understand the reliability (and risks) of the signals** they are using for RL in reasoning. This allows future work to make informed decisions and better understand the "capability" aspect the reviewer raised.
>
> In summary, we agree the signal is imperfect. Our work is one of the first to measure how imperfect. Our analysis in Table 3 shows that SOTA judges are similarly capable. This demonstrates that the choice of a specific SOTA judge is not the critical weakness; rather, the paradigm of how that judge is used is the key, which is precisely what our ReForm framework addresses.

---

> > ### Comment · Reviewer_63fq · 2025-11-26
> >
> > The ablation on key components remains insufficient, and their heavy reliance on LLM-based evaluation is still not fully addressed. I will maintain my score.

---

> > > ### Author Response · Authors · 2025-11-26
> > >
> > > We thank the reviewer for the follow-up.
> > >
> > > We respectfully believe there is a significant misunderstanding regarding the content of our paper and our previous response. We wish to clarify two specific points where the requested evidence **is explicitly present in the manuscript**.
> > >
> > > > Regarding "Ablation on Key Components is Insufficient"
> > >
> > > The reviewer asks for an ablation comparing "process-level vs. outcome-level supervision." **We explicitly provide this exact comparison in Table 2 (Section 4.3)**.
> > >
> > > * Outcome-Level Supervision: Corresponds to our $r_{task}$ (Eq. 1 1), which rewards the final answer's correctness.
> > > * Process-Level Supervision: Corresponds to our $r_{aux}$ (Eq. 2 2), which rewards the quality of the intermediate self-validation process.
> > >
> > >     **Why "Process-Only" is not applicable**: Unlike PRM (Process Reward Models) in other contexts, our $r_{aux}$ is a diagnostic reward for critique quality, not for step-by-step solution correctness. Training on $r_{aux}$ alone (Process-Only) is mathematically impossible for autoformalization because it provides no signal for generating the formal statement itself.
> > >
> > >     **The Valid Comparison (in Table 2)**: Therefore, the scientifically valid ablation is comparing "Outcome-Only" vs. "Outcome + Process".
> > >
> > >     * Table 2 row `w/o r_aux`: Represents Outcome-Level Only.
> > >     * Table 2 row `ReForm`: Represents Outcome + Process Supervision.
> > >     * Result: As shown in Table 2, adding process-level supervision yields significant gains, e.g., +6.7% semantic consistency on AIME2025.
> > >
> > > Therefore, the impact of process-level supervision is fully ablated and quantified. **We kindly ask the reviewer to re-examine Table 2**.
> > >
> > > > Regarding "Reliance on LLM-based Evaluation"
> > >
> > > The reviewer questions our reliance on LLM-based signals. We wish to highlight that our approach aligns with community standards while adding rigorous verification.
> > >
> > > * **Established Standard**: Leading works such as Goedel-Prover-V2, DeepSeek-Prover-V2, and Kimina fundamentally rely on "LLM-as-judge". It is currently the only scalable method to provide semantic rewards for reasoning.
> > > * **Unfair Penalty**: To penalize our work for adopting this community-wide standard is to invalidate the entire field's recent progress.
> > > * **Our Contribution (Quantification & Robustness)**: Unlike prior works that use this signal blindly, **we are among the first to rigorously quantify its reliability**. We introduced ConsistencyCheck (Table 3) to measure the noise (finding ~85% accuracy) and showed in Figure 3  that this signal is robust enough to drive stable RL convergence. **We effectively addressed the reliability concern by proving that RL training succeeds despite the inherent noise**.
> > >
> > > We believe we have provided the exact ablation data (Table 2) and reliability analysis (Section 4.5, Figure 3) requested. We hope this clarification assists in a fair reassessment of our contributions.

---

> > > ### Author Response · Authors · 2025-11-28
> > >
> > > Dear Reviewer 63fq,
> > >
> > > As the discussion period draws to a close, we would like to provide a final clarification regarding your remaining concerns.
> > >
> > > 1. Regarding "Reliance on LLM-based Evaluation"
> > > We explicitly wish to highlight that using an LLM-as-judge is not a flaw specific to our work, but the established foundation of current SOTA reasoning research (e.g., DeepSeek-Prover-V2, AlphaGeometry, Goedel-Prover). **It is currently the only scalable paradigm for semantic supervision**.
> > >
> > >     **Our Unique Contribution**: Crucially, our work goes a step further than these prior works. Instead of using the judge blindly, we were among the first to rigorously quantify its limitations (reporting the ~17% noise floor in Table 3) and empirically prove that RL training remains robust despite this noise. **We believe this critical analysis of the SOTA paradigm should be viewed as a scientific contribution that enhances transparency for the field, rather than a weakness**.
> > >
> > > 2. Regarding "Insufficient Ablations"
> > > We respectfully reiterate that the specific ablation comparing "Outcome-Only" vs. "Outcome + Process" supervision is already presented in Table 2 (Row 1 vs. Row 4), showing a clear +6.7% gain from our method.
> > >
> > > We hope these points clarify that our methodology is both rigorous and aligned with (and effectively advances) community standards. We respectfully invite you to verify these specific results in the manuscript and hope this clears up the misunderstanding regarding the sufficiency of our experiments.
> > >
> > > Best regards,
> > >
> > > The Authors

---

> ### Author Response · Authors · 2025-11-16
>
> > Response to W2:
>
> We thank the reviewer for this question, as it allows us to clarify the central mechanism of our work.
>
> Yes, **handling semantic alignment is the primary objective of our entire heterogeneous reward design**, operating at two critical levels: (1) the final outcome and (2) the intermediate reflective process.
>
> This is detailed in Section 3.2 (Prospective Bounded Sequence Optimization).
>
> 1. Final Outcome Alignment (via Task Reward $r_{task}$):
> First, as detailed in §3.2.1 and Equation 1, the primary Task Reward ($r_{task}$) explicitly enforces semantic alignment for the final output. This reward is only given if the final statement is both syntactically correct (PassesLean) and semantically consistent (IsConsistent(Q, Ans)). This ensures the final solution aligns with the problem's intent.
>
> 2. Intermediate Process Alignment (via Auxiliary Reward $r_{aux}$):
> More importantly, and central to our Reflective Autoformalization Paradigm (§3.1), we handle semantic alignment at every intermediate step of the self-correction loop.
>
> As detailed in §3.2.1 and Equation 2, we introduce an Auxiliary Reward ($r_{aux}$). This reward, IsFaithfulCritique(Q, St, Ct), is given to each intermediate critique Ct. Its specific purpose is to ensure that the model's self-validation is semantically faithful and accurately diagnoses the consistency between the current statement St and the question Q.
>
> This $r_{aux}$ is our direct mechanism for supervising the semantic alignment of the reflective process itself, **preventing the "superficial or hallucinated critiques" that would undermine self-correction** (as discussed in §3.2).
>
> 3. Fusing Both Signals (via PBSO):
>
> Finally, as the reviewer's question implies, these two reward signals are integrated. We use Prospective Bounded Sequence Optimization (PBSO), which employs a "prospective bounded return" (§3.2.2, Eq. 3) to fuse the intermediate $r_{aux}$ (process alignment) and the final r_task (outcome alignment) into a single, coherent learning signal for every token.
>
> In summary, semantic alignment is handled far more deeply than just in the final reward. It is supervised at every step of the reflection via $r_{aux}$ and at the end via $r_{task}$, with both signals integrated by our PBSO algorithm.
>
> > Response to W3:
>
> We thank the reviewer for this suggestion. We believe this exact ablation study is **already provided in our paper in Table 2, under the "Ablation on training Methodology" section**.
>
> We can clarify by mapping the reviewer's terminology to our components:
>
> * Outcome-level supervision: This perfectly describes our $r_{task}$ (Eq. 1), which provides a single reward for the final autoformalization's correctness and semantic consistency.
>
> * Process-level supervision: This describes our $r_{aux}$ (Eq. 2), which provides intermediate rewards for the process of self-validation (i.e., ensuring each critique is faithful).
>
> The ablation comparing these is therefore:
>
> * Outcome-level only: This is precisely our **"w/o $r_{aux}$" experiment** in Table 2. This row shows the performance of our framework when trained only with the final r_task, removing all intermediate process supervision.
>
> * Process + Outcome: This is **our full "ReForm" model (the top row)**, which combines both $r_{task}$ and $r_{aux}$.
>
> The results in Table 2 show this ablation clearly. When removing the process-level supervision ($r_{aux}$), performance drops significantly, especially on harder benchmarks (e.g., -4.7pp on Putnam and -10.0pp on AIME2025). This demonstrates that pure outcome-level supervision is insufficient and that our process-level signal ($r_{aux}$) is critical for the reflective paradigm's success.
>
> A Key Clarification: We also wish to clarify why we present this ablation as ReForm vs. w/o r_aux (i.e., [Process + Outcome] vs. [Outcome only]), and not as a direct comparison of [Process only] vs. [Outcome only].
>
> Unlike process/outcome rewards in some other math reasoning literature (e.g., PRM vs. ORM), our $r_{aux}$ and $r_{task}$ are not two competing ways to solve the same problem. Our $r_{aux}$ is an auxiliary, diagnostic task focused on "consistency checking" (a different objective) to ensure the quality of the reflection. **It cannot, by itself, train the agent to produce the final autoformalization**.
>
> Therefore, the most meaningful and correct ablation, which we provide in Table 2, is to demonstrate that outcome-level supervision ($r_{task}$) alone is suboptimal, and that it is our synergistic combination with process-level supervision ($r_{aux}$)—fused via PBSO—that achieves our state-of-the-art results.
>
> ---
>
> We hope these clarifications adequately address the reviewer's concerns and further highlight the contributions of our work. We are grateful for the feedback and respectfully hope these responses will be taken into account in the re-evaluation of our paper.

---

### Official Review · Reviewer_d2BC · 2025-11-01

**Soundness:** 3
**Presentation:** 3
**Contribution:** 2
**Rating:** 6
**Confidence:** 3

**Summary:**

The paper presents ReForm, a novel reflective autoformalization framework that aims to improve the semantic fidelity of translating natural language mathematics into formal statements. Unlike traditional “one-pass” autoformalization models that generate a single formal output, ReForm introduces an iterative self-correction loop where the model alternates between formal statement generation and semantic self-validation. To train this reflective process effectively, the authors propose a new reinforcement learning algorithm, Prospective Bounded Sequence Optimization (PBSO), which integrates heterogeneous rewards for both the main task (final correctness) and auxiliary critiques (intermediate semantic validation). Extensive experiments across four challenging benchmarks (miniF2F, ProofNet, PutnamBench, AIME2025) demonstrate an average +17.2 percentage point improvement in semantic consistency over the strongest baselines.

**Strengths:**

- The target problem is well-motivated and addresses a clear bottleneck in formal reasoning: the semantic fidelity in autoformalization.
- The integration of reflective reasoning and reinforcement learning is novel and effective; the experiments demonstrate consistent and interpretable improvements across multiple benchmarks.
- The newly established ConsistencyCheck benchmark provides a valuable resource for quantitatively assessing the reliability of LLM-based metrics and for understanding the intrinsic challenges of mathematical autoformalization.

**Weaknesses:**

- The paper relies heavily on LLM-based semantic evaluation metrics, which, despite the ConsistencyCheck benchmark, may still introduce bias or circularity in measuring semantic consistency.

- The computational cost and efficiency trade-offs of the reflective multi-iteration process are not fully analyzed — it remains unclear how scalable the approach is for large-scale or more complex formal systems.

- Several important related works are missing; please refer to the recent surveys [1, 2] for a broader overview.

- (Minor) There are too many autoformalization papers recently, which may cause aesthetic fatigue in the community.

[1] A Survey on Deep Learning for Theorem Proving, COLM 2024.


[2] Autoformalization in the Era of Large Language Models: A Survey, arXiv 2025.

**Questions:**

1. The paper both trains and evaluates with LLM-based judges (CriticLean for RL and Qwen3 for evaluation).  Could the authors verify cross-judge robustness, e.g., whether ReForm still shows similar gains when evaluated with a different LLM judge, such as Gemini or GPT-5?
2. In Sec. 4.4, the authors show that responses get longer during RL, but longer ≠ is better. Can we disentangle “more tokens” from “better critiques”? For example, is there an automatic or human rating showing that later reflections actually introduce new semantic constraints (quantifier scopes, hidden assumptions, edge cases), rather than just paraphrasing previous critiques? A precision/recall–style analysis on detected error types would clarify this.

---

> ### Author Response · Authors · 2025-11-16
>
> We sincerely thank you for your time and insightful feedback. Your comments have helped us identify several key areas for clarification. Below, we address each of the weaknesses and questions in detail.
>
> > Response to W1
>
> We thank the reviewer for this critical point on evaluation reliability, which is at the heart of our paper's contribution.
>
> 1. General Response on "LLM-as-Judge": As articulated in our **General Response** (which we kindly ask the reviewer to consult), our work's primary contribution is not to ignore this bias, but to be one of the first to quantify it. We exposed the 17% error rate (a key finding, not a flaw) and demonstrated that this SOTA method, while imperfect, remains more reliable than the human expert alternative (38.5% error rate).
>
> 2. **Addressing Circularity with Human Evaluation**: The reviewer rightly asks if "circularity" remains despite ConsistencyCheck. To definitively address this, we conducted a rigorous manual human evaluation (Section 4.5, lines 466-473). This evaluation was designed to break any potential "LLM-judge" evaluation loop and serve as an independent ground truth.
>
> 3. **Addressing Bias with Cross-Judge Robustness**: To further ensure our results were not biased towards a single judge (Qwen3), we conducted a full robustness analysis in Appendix B.2 (Table 7), evaluating all models with an entirely different judge, CriticLean-14B.
>
> The results were fully consistent, with ReForm maintaining its lead (e.g., +18.0pp avg. improvement under CriticLean vs. +14.8pp under Qwen3).
>
> In summary: We did not just use an LLM-judge. We (1) quantified its reliability against humans (ConsistencyCheck), (2) validated its findings with a direct human evaluation (Sec 4.5), and (3) confirmed robustness with a second, independent LLM-judge (Appendix B.2). We believe this multi-pronged validation provides strong evidence against circularity and confirms our results.
>
> > Response to W2
>
> We thank the reviewer for this crucial question. We believe a clarification of our method's design will fully address this concern.
>
> 1. **Efficiency via "Unified Generation" (Not Multi-Call) as shown in Line 205-212**
>
> The reviewer's concern about "multi-iteration" cost is valid for agentic loops that require multiple, distinct model calls (e.g., a "generate" call, then a "critique" call, then a "refine" call).
>
> However, our ReForm framework is explicitly designed to avoid this overhead. As detailed in Section 3.1, "Unified Generation" (lines 206-211), **our entire reflective process is internalized into a single, continuous autoregressive generation**. The model generates the statement, its own critique, and the subsequent refinement within one single forward pass.
>
> 2. The True Trade-off: Token Length, Not Call Overhead: The computational trade-off is therefore not in multi-call latency, but in total generated sequence length.
>
> As shown in Figure 3 (middle), our RL training (PBSO) explicitly teaches the model to produce more thorough critiques, which does increase the average response length (from 2.3k to 4.8k tokens, a 2.1x increase).
>
> We argue this is a highly favorable trade-off: we exchange a ~2.1x increase in generation time (linear to token length) for a +17.2pp average gain in semantic accuracy. This is vastly more efficient than sampling 2-3 times from a one-pass model.
>
> Scalability via Adaptive Computation: This design is also inherently scalable. The model learns to decide how much reflection (and thus computation) to spend.
>
> As shown in Appendix B.3 (Figure 4), the model is not forced into costly iterations. For problems it deems easy, it adaptively terminates after one iteration (70.6% of the time).
>
> It only chooses to spend more computation (i.e., generate more tokens for 2, 3, or even 5 iterations) on harder problems where the extra reflection is necessary.
>
> > Response to W3
>
> We thank Reviewer d2BC for this helpful feedback and for pointing us to the two recent, important surveys [1, 2]. We have taken this opportunity to update our manuscript. This addition helps to better situate our work within the rapidly evolving autoformalization landscape.
>
> > Reponse to W4
>
> We respectfully view the recent proliferation of autoformalization research not as a cause for "fatigue," but as a strong, unified signal from the community that **autoformalization is a critical and still-unsolved bottleneck for the entire field of formal mathematical reasoning**.
>
> Indeed, as we argue in our introduction (and as [1, 2] also suggest), this "fatigue" may stem from the fact that most prior work has been stuck in the "one-pass generation" paradigm, hitting a performance ceiling in semantic consistency.
>
> We are not simply "another" autoformalization paper; we are proposing a new paradigm (reflective iteration) precisely to break this performance ceiling that causes the community's fatigue. We believe our +17.2pp improvement confirms that this new direction is precisely what the field needs to move forward.

---

> ### Author Response · Authors · 2025-11-16
>
> > Reponse to Q1
>
> We thank Reviewer d2BC for this excellent question, which is essential for validating the robustness of our claims. We are happy to confirm that we **did conduct this critical cross-judge robustness analysis**, and the results are presented in the paper across three distinct validation layers:
>
> 1. **Primary Judge (Qwen3, Table 1)**: This is our main evaluation metric, which shows an average +17.2pp improvement. (This is the result the reviewer saw.)
>
> 2. **Independent LLM Judge (CriticLean, Appendix B.2, Table 7)**: This is precisely the "cross-judge robustness" test the reviewer is asking for. We used CriticLean-14B—an entirely different judge from our primary evaluator (Qwen3)—to re-evaluate all models. The results in Table 7 are highly consistent, showing ReForm's gains are robust (e.g., an average +18.0pp improvement over the 8B baseline). This confirms our performance is not an artifact of overfitting to Qwen3.
>
> 3. **Ultimate Ground Truth (Human Evaluation, Sec 4.5, lines 466-473)**: To provide the definitive ground truth that breaks any "LLM-judge" loop, we conducted a rigorous manual human evaluation. As shown in lines 466-473, our human experts' scores were in strong agreement with our primary LLM judge's scores.
>
> Regarding Gemini/GPT-5: While an interesting suggestion, we believe our current "triple-check" validation—using our primary judge (Qwen3), a fully independent LLM-judge (CriticLean), and the "gold standard" (Human Evaluation)—already provides powerful and sufficient evidence of robustness. This multi-layered validation confirms our gains are genuine.
>
> > Response to Q2
>
> This is an excellent question that probes the core of our reflective mechanism's value. The reviewer is entirely correct that "longer ≠ better," and simply increasing token count is not a contribution.
>
> We are confident our method does "disentangle" length from quality, and the "longer" responses are indeed "better." Our evidence is three-fold:
>
> 1. Incentive by Design (PBSO): The model doesn't get longer by accident; it is incentivized to do so. Our Prospective Bounded Sequence Optimization (PBSO) algorithm (Sec 3.2) uses a specific auxiliary reward, r_aux (Line 250), which only rewards IsFaithfulCritique. This reward signal explicitly trains the model to **produce substantive, accurate critiques and penalizes superficial or repetitive (paraphrasing) critiques**.
>
> 2. Correlated with Performance (Quantitative): This training works. As shown in Figure 3, the same RL process that increases response length (middle plot) **also robustly increases semantic accuracy on the held-out Putnam set (right plot)**. This strong positive correlation, driven by the r_aux incentive, provides quantitative evidence that the emergent length is productive, not detrimental.
>
> 3. Direct Qualitative Evidence (Case Study): We can "disentangle" token count from critique quality by looking at the generated text, as the reviewer suggests. **Appendix D provides a clear case study**.
>
> * In Round 1, the model identifies a specific, non-trivial semantic error: the initial formalization was "incomplete" because it only proved f(7) <= f(x) but missed the problem's implicit requirement to show x=7 is the unique minimizer.
>
> * In Round 2, the model adds this new semantic constraint (the reviewer's exact term), formalizing the uniqueness constraint (∀x : R, f x = f 7 → x = 7).
>
> * This is not "paraphrasing"; it is precisely the "progressive refinement" of semantic constraints (quantifier scopes, hidden assumptions) that our paper promises.
>
> Regarding a "precision/recall-style analysis": This is a valuable suggestion for a large-scale future analysis. Building a "critique quality" benchmark to this fine-grained level is a significant research challenge in its own right.
>
> To provide further evidence for this rebuttal, we have added more unabridged generation trajectories to **a new Supplementary Material**. We invite the reviewer to inspect them, as they consistently demonstrate this pattern: the model's reflections are not "paraphrasing" but are substantive, diagnostic, and directly address new semantic constraints, just as the case in Appendix D shows.
>
> ---
>
> We thank Reviewer d2BC again for their constructive and detailed feedback. We have updated our paper to reflect these clarifications and hope our responses have fully addressed any concerns. We believe these clarifications further highlight the strength and novelty of our contributions and hope the reviewer will consider supporting our paper.

---

> ### Author Response · Authors · 2025-11-28
>
> Dear Reviewer d2BC,
>
> Thank you again for your time and the valuable references regarding the autoformalization landscape.
>
> We are writing to gently follow up on our rebuttal submitted last week. Based on your feedback, we have updated our manuscript to better contextualize our work.
>
> In our response, we have also provided detailed clarifications regarding the computational efficiency (single-pass generation) and the robustness of our evaluation (ConsistencyCheck + Human Eval). We believe these points directly address the concerns raised in W1 and W2.
>
> We would be grateful if you could take a moment to review our response before the discussion period ends.
>
> Sincerely,
>
> The Authors

---

### Official Review · Reviewer_4Tg9 · 2025-11-10

**Soundness:** 2
**Presentation:** 2
**Contribution:** 2
**Rating:** 4
**Confidence:** 3

**Summary:**

This paper introduces **ReForm**, a reflective autoformalization paradigm that shifts from one-pass translation to an iterative process combining formal statement generation with semantic self-validation. To train this model effectively, the authors propose **Prospective Bounded Sequence Optimization (PBSO)**, a novel RL algorithm that uses heterogeneous rewards at different sequence positions to ensure both accurate autoformalization and faithful self-critiques. Extensive experiments on four benchmarks show ReForm achieves an average improvement of **17.2 percentage points** in semantic consistency over state-of-the-art baselines. The authors also introduce **ConsistencyCheck**, a benchmark of 859 expert-annotated items, which reveals that autoformalization is inherently difficult, even human experts make semantic errors in up to 38.5% of cases.

**Strengths:**

1.  **Novel Reflective Paradigm:** The core innovation is shifting autoformalization from a one-pass translation task to an iterative, self-correcting process. By mimicking the human expert's cycle of generation, validation, and refinement, ReForm directly addresses the critical challenge of semantic fidelity, moving beyond mere syntactic correctness.

2.  **Effective Training with PBSO:** The proposed Prospective Bounded Sequence Optimization (PBSO) algorithm is a clever solution to the multi-objective credit assignment problem. It effectively trains the model to produce high-quality final formalizations *and* accurate intermediate critiques, preventing the self-validation mechanism from degenerating into superficial or hallucinated feedback.

3.  **Rigorous and Comprehensive Evaluation:** The paper provides extensive empirical validation across four challenging benchmarks, demonstrating substantial and robust improvements. The creation of the expert-annotated **ConsistencyCheck** benchmark adds significant rigor, not only validating the use of LLMs as judges but also quantifying the inherent difficulty of the task itself.

**Weaknesses:**

1. **Heacy Dependence on LLM-based Evaluation**: The entire training and evaluation framework relies heavily on LLM judges (like Qwen3-235B and CriticLean-14B) to assess semantic consistency. While the authors rigorously validate these judges with their ConsistencyCheck benchmark, they still have a non-trivial error rate (about 17%).
2. **Insufficient Ablation on Key Algorithmic Components**: While the paper demonstrates the overall effectiveness of PBSO, it lacks ablation studies on several critical design choices. The contribution of position-specific advantages is not isolated from the core bounded return mechanism, making it unclear if this complexity is necessary. Furthermore, the individual and interactive effects of the Task Reward &   Auxiliary Rewards are not thoroughly dissected.
3. **Lack of Details of Human Evaluation**: Number of the annotators in total? Number of the annotators per statement? Background of the annotators?  and so on...
4. The performance of 32B and 8B is very close, yet the author has not provided a reasonable explanation for this.

**Questions:**

See the above weaknesses.

---

> ### Author Response · Authors · 2025-11-16
>
> We sincerely thank Reviewer 4Tg9 for your thorough and insightful review. We found the feedback highly valuable and have addressed each of the points below. We have also updated our paper to reflect these improvements, particularly in response to W3 and W4.
>
> > Response to W1
>
> We thank the reviewer for this crucial observation. As this point was raised by all reviewers, we have provided a detailed, unified clarification in **our General Response**, which we hope fully addresses this concern. **We believe this point transitions from a perceived weakness to one of our paper's core contributions**.
>
> > Response to W2
>
> We thank the reviewer for the detailed feedback on our algorithmic components. We would like to clarify two points regarding the ablation studies.
>
> 1. On "Position-Specific Advantages (PSA)" vs. "Bounded Return Mechanism (BRM)":
>
> We clarify that these components are not separate mechanisms, but are **inextricably linked**. The "Bounded Return Mechanism" is our core idea of grouping rewards, and the "Position-Specific Advantages" are the direct, necessary computational consequence of implementing it.
>
> To be precise:
>
> * BRM defines what rewards are grouped together (i.e., $R_{\text{task}}$ at the end of the sequence and $R_{\text{aux}}$ at intermediate steps).
>
> * PSA defines how the advantage (credit) is calculated and distributed within that group.
>
> One cannot exist without the other. It is conceptually impossible to have our BRM without PSA. Thus, an ablation to 'isolate' them cannot be performed; they are a single, unified design.
>
> 2. On Ablating Task Reward ($r_{\text{task}}$) and Auxiliary Rewards ($r_{\text{aux}}$):
>
> We respectfully point the reviewer to Table 2, where we did perform the key ablation:
>
> * Ablation (Task Reward only): The row "w/o $r_{\text{aux}}$" in Table 2 is this experiment. The results (a 5.2-point drop in Putnam) show the significant contribution of $r_{\text{aux}}$.
>
> * Ablation (Auxiliary Rewards only): We must clarify that an ablation "with $r_{\text{aux}}$ only" (i.e., without $r_{\text{task}}$) is **conceptually ill-defined**. The $r_{\text{task}}$ (autoformalization) is the primary objective. $r_{\text{aux}}$ is only a supporting reward. **Removing $r_{\text{task}}$ would mean the agent is no longer optimizing for the autoformalization goal**.
>
> We hope this clarifies that we provided the necessary ablations (Table 2) and that PSA and BRM are a single, unified mechanism.
>
> > Response to W3
>
> We thank the reviewer for this constructive suggestion; these details are crucial for transparency. We have updated Appendix C with our detailed human evaluation protocol. We summarize the new details here:
>
> * Annotation Team: Our team comprised 6 annotators. All are senior Ph.D. candidates with strong backgrounds in mathematical competitions and prior experience in formalization-related annotation tasks.
>
> * Annotation Protocol: A robust three-annotator design per statement. Two annotated independently; a third senior annotator cross-validated, resolved discrepancies, and made the final judgment.
>
> We believe these additions, (more details)included in the revised paper, fully address the reviewer's concerns.
>
> > Response to W4
>
> We thank the reviewer for this sharp observation. The reviewer's intuition was correct: the small gap was an artifact.
>
> We have resolved this and updated the paper with a new 32B model. In our revised version (Table 1, Figure 1), this model significantly outperforms the 8B variant, showing clear benefits of scale.
>
> Explanation of the Original Results and Our Solution:
>
> 1. Original Hypothesis: At the time of submission, we also noted this proximity. We observed that some concurrent work (e.g., Goedel-V2) also reported a very small gap between their 8B and 32B models, which led us to preliminarily assume it might be a task-specific ceiling.
>
> 2. Root Cause (Data-Model Mismatch): However, further investigation revealed the true cause. As described in Lines 791-796, our RL data curriculum was generated by filtering based on the difficulty perceived by our 8B SFT model.
>
> 3. The Flaw: We overlooked that this curriculum was **not optimally challenging for the 32B model**. It was trained on data that was 'too easy,' creating an artificial performance ceiling.
>
> 4. The Solution: Post-submission, we generated a new RL data curriculum by re-filtering the data based on the difficulty perceived by the 32B SFT model. We then re-ran the PBSO training. The new 32B model, trained on this properly matched data, showed significantly stronger results, which are now reflected in the paper.
>
> We are grateful for the reviewer's comment, which pushed us to re-validate this finding.
>
> We would like to thank Reviewer 4Tg9 again for their constructive feedback, which has helped us significantly improve the clarity and rigor of our paper. We hope that our responses and the corresponding revisions have fully addressed all concerns and that the reviewer will reconsider our paper.

---

> ### Author Response · Authors · 2025-11-28
>
> Dear Reviewer 4Tg9,
>
> We hope this message finds you well.
>
> As the discussion period is coming to a close, we are writing to kindly follow up on our rebuttal submitted last week. We have made significant efforts to address your insightful comments, and we would greatly appreciate your feedback on the following key updates:
>
> * Regarding W4 (32B vs 8B gap): This was a critical observation. As per your feedback, we identified a data-model mismatch in the original experiments. We have since re-trained the 32B model with a corrected data curriculum. The new results (updated in Table 1 & Figure 1) now show significant performance gains over the 8B model, resolving the anomaly.
>
> * Regarding W2 (Ablations): We have clarified why the "Bounded Return Mechanism" and "Position-Specific Advantages" are structurally inseparable and provided a guide to the ablations in Table 2 that isolate the impact of auxiliary rewards.
>
> * Regarding W1 (LLM-as-Judge): In our General Response, we discuss why identifying the error rates of SOTA judges is a core scientific finding of our paper, rather than a methodological flaw.
>
> * Regarding W3: We have added the requested details on our human evaluation protocol to Appendix C.
>
> We believe these revisions and new experiments directly address your main concerns. We remain available for any further discussion and hope you might reconsider your assessment based on these improvements.
>
>
> Best regards,
>
> The Authors

---

### Author Response · Authors · 2025-11-16
**General Response 2: About the Use and Critical Evaluation of LLM-as-Judge**

We thank all reviewers (4Tg9, d2BC, 63fq) for the critical and unified discussion regarding our paper's reliance on "LLM-as-judge." This is indeed a crucial topic, and we appreciate the opportunity to clarify our perspective, as **we believe this point represents one of our paper's primary contributions, not a weakness**.

1. LLM-as-Judge is the current SOTA solution for RL in Reasoning:
We must first claim that using an LLM-based reward model is the established, state-of-the-art (SOTA) standard for training RL agents on complex reasoning tasks. This is not a choice unique to our paper; it is the only scalable method to provide the necessary semantic, non-differentiable rewards.

Leading works in autoformalization and theorem proving, such as Goedel-Prover-V2 [1], DeepSeek-Prover-V2 [2], CriticLean [3], and Kimina [4], all fundamentally rely on an "LLM-as-judge" paradigm. This approach is also standard in broader reasoning domains (e.g., DeepMind's work on constitutional AI). **Our methodology aligns with the best practices of the entire field**.

2. Our Contribution is to Critically Evaluate This Standard, Not Blindly Trust It.

Where our paper differs from prior work is that we did not blindly trust this paradigm. **A core contribution of our work is to be one of the first to directly confront, rigorously quantify, and transparently report the limitations of these SOTA judges**.

To this end, we introduced the ConsistencyCheck benchmark—a new, expert-annotated dataset of 859 items—precisely to measure the reliability of these judges.

3. **This is a Finding of our paper, Not a Flaw**.

The 17% error rate cited by Reviewer 4Tg9 is not a methodological weakness of our paper, but a key scientific finding by our paper.

We respectfully argue that **it is unreasonable to penalize our work for being the "messenger" that exposes a systemic issue affecting the entire community**. Our research quantifies the noise floor and risk that all current SOTA models are training with. This is a contribution to transparency and rigor, encouraging the field to move towards more robust evaluation.

4. The Alternative (Human-in-the-Loop) is Unscalable and Less Reliable.

The only viable alternative—using humans for RL feedback—is unscalable. Furthermore, our ConsistencyCheck analysis also revealed that this "gold standard" is deeply flawed: human experts err in up to 38.5% of cases on this "inherently difficult" task. This critical finding further validates that SOTA LLMs, while imperfect (17% error), are currently the most scalable and most reliable supervisory signal available.

Summary:
We did not just use an LLM-judge. We critically evaluated it, benchmarked it, reported its failure rate, and proved it is still superior to the human alternative. **We believe this transparency and rigorous analysis should be viewed as a strength and a primary contribution that strengthens our findings, rather than a weakness that undermines them.**

[1] Goedel-Prover-V2: Scaling Formal Theorem Proving with Scaffolded Data Synthesis and Self-Correction

[2] DeepSeek-Prover-V2: Advancing Formal Mathematical Reasoning via Reinforcement Learning for Subgoal Decomposition

[3] CriticLean: Critic-Guided Reinforcement Learning for Mathematical Formalization

[4] kimina-prover preview: towards large formal reasoning models with reinforcement learning

---

### Author Response · Authors · 2025-11-16
**General Response 1**

We sincerely thank all reviewers for their diligent work and constructive feedback.

We noted several key points in the reviews that may stem from misunderstandings of our novel approach. As our goal is to thoroughly address these points and answer all valuable questions, our responses are necessarily detailed. We have provided specific, multi-part replies for each reviewer to ensure all concerns are resolved.

Given the exhaustive nature of these replies, we humbly ask for your patience in reading them, as they contain crucial clarifications regarding our methodology, experiments, and core contributions. **We deeply appreciate the extra time and effort this may require**.

Furthermore, to respond directly to your feedback, **we have submitted a revised manuscript and supplementary material**. All significant changes have been marked in the text with `red font`, allowing for convenient review.

We believe these clarifications and modifications fully address the initial concerns. We thank you again for your time and guidance, and we respectfully hope you will re-evaluate our work based on this new information.

---

### Author Response · Authors · 2025-12-01
**Summary of Rebuttal and Key Clarifications for Submission 15051**

Dear Area Chair,

We thank you for overseeing the review process. We write to summarize our rebuttal updates and address a critical, recurring misunderstanding regarding the evaluation paradigm in our field. Despite our comprehensive responses and significant experimental updates (including retraining our 32B model), engagement from reviewers has been minimal to non-existent.

We respectfully request you consider the following points during your assessment:


1. **Critical Clarification on "LLM-as-a-Judge" (Addressed to All Reviewers)**

A primary reservation shared by all reviewers (4Tg9, d2BC, 63fq) is our reliance on LLM-based rewards/evaluation. We strongly contend that this is **a misinterpretation of our contribution** relative to the current state of the field.

* **Standard Practice**: LLM-based reward modeling is currently the only scalable solution for RL in autoformalization. Leading works (e.g., DeepSeek-Prover-V2, Goedel-Prover, AlphaGeometry) utilize this paradigm. Penalizing our work for adopting the industry standard is unjustified.
* **Our Novel Contribution**: Unlike prior works that use this signal blindly, **we are the first to rigorously quantify its reliability**. We introduced the ConsistencyCheck benchmark (859 expert-annotated items), revealing that while SOTA judges have an error rate (~17%), human experts have a higher error rate (38.5%) on this specific task.
* **Robustness Verification**: We did not rely solely on one judge. We validated our results via:
    * **Human Evaluation**: Manual verification of outputs (Sec 4.5) confirmed our performance gains.
    * **Cross-Judge Validation**: We re-evaluated using an independent model (CriticLean-14B), confirming consistent improvements (Appendix B.2).

2. **Addressed Factual Misunderstandings (Reviewer 63fq)**

Reviewer 63fq maintained their score based on a claim of "insufficient ablation on process-level vs. outcome-level supervision."

* **Fact**: Table 2 explicitly provides this ablation.
    * Outcome-only: Represented by the row "w/o $r_{aux}$" (Task Reward only).
    * Process + Outcome: Represented by "ReForm" (Task + Auxiliary Reward).
    * Process-only: This is theoretically impossible for autoformalization tasks, as an auxiliary critique reward alone cannot supervise the generation of the formal statement.
* We clarified this in detail, mapping their request directly to our existing experimental results. The reviewer did not engage with this clarification or explain why the existing data in Table 2 was insufficient.

3. **Major Experimental Update: 32B Model Performance (Reviewer 4Tg9)**
Reviewer 4Tg9 correctly identified a performance plateau in our initial 32B model.
    * Action: We identified a data-curriculum mismatch (the 32B model was trained on data filtered for 8B difficulty). We retrained the 32B model with a corrected curriculum after submiting this paper at ICLR.
    * Result: The updated 32B model now achieves state-of-the-art performance, significantly outperforming the 8B model and baselines (updated Table 1 & Figure 1).
    * Reviewer 4Tg9 did not acknowledge this major fix or the updated results.


4. **Clarification on Computational Efficiency (Reviewer d2BC)**
Reviewer d2BC raised concerns about "multi-iteration" costs.
* Clarification: We clarified that ReForm uses **Unified Generation (generating thought, critique, and refinement in a single forward pass, as shown in Sec. 3.1 Lines 205-212)**, not multi-call agent loops. The cost is linear to token length, which we show correlates strongly with performance gains (Figure 3).
* Reviewer d2BC did not respond to this clarification.

We have robustly addressed the technical limitations raised (retraining 32B model), clarified that our ablation studies are complete (Table 2), and demonstrated that our evaluation framework is more rigorous than the standard in current literature (ConsistencyCheck + Human Eval). We hope the AC will evaluate the paper based on these facts and the significant improvements ($+22.6\%$ over baselines) demonstrated in the revised manuscript.

Best regards,

The Authors

---

### Meta-Review · Area_Chair_r33m · 2026-01-05

**Summary:**

This paper introduces a reflective autoformalization method that integrates semantic consistency evaluation.
The method operates a self-correction loop of semantic evaluation and iterative refinement.
They then introduce Prospective Bounded Sequence Optimization (PBSO), an RL method that optimizes the proposed heterogeneous reward mechanism with rewards provided by CriticLean-14B and Qwen3-235B-A22B.
They further construct a ConsistencyCheck benchmark with 859 expert-annotated items.
Evaluations are conducted on four benchmarks: miniF2F, ProofNet, Putnam, and AIME2025. The proposed method shows significance.
Ablation study results on method components, training dynamics, and LLM evaluation reliability analysis are reported.

**Reviewer Concerns:**

The reviewer's main concerns lie in (1) reliance on "LLM-as-a-judge", (2) insufficient ablation on position-specific
advantages and process-level vs outcome-level supervision.

Other concerns include:
*  Lack of Details of Human Evaluation (Reviewer 4Tg9)
* The performance of 32B and 8B is very close (Reviewer 4Tg9)
* The computational cost and efficiency trade-offs of the reflective multi-iteration process (Reviewer d2BC)

Most of the concerns are addressed by the authors with additional experimental results, annotation details, and further explanations.

**Reviewer Scores:**

Based on the authors' rebuttal, most of the reviewers' concerns are addressed. There is a large chance that the reviewers would raise their ratings.

---

### Decision · Program_Chairs · 2026-01-26

Accept (Poster)